# Polypharmacology-based kinome screen identifies new regulators of KSHV reactivation

Annabel T. Olson[1], Yuqi Kang[1], Anushka M. Ladha[2], Songli Zhu[1], Chuan Bian Lim[1], Behnam Nabet[1], Michael Lagunoff[2], Taranjit S. Gujral[1,3]*, Adam P. Geballe[1,2,4,5]*

1 Division of Human Biology, Fred Hutchinson Cancer Center, Seattle, Washington, United States of America, 2 Department of Microbiology, University of Washington, Seattle, Washington, United States of America, 3 Department of Pharmacology, University of Washington, Seattle, Washington, United States of America, 4 Division of Clinical Research, Fred Hutchinson Cancer Center, Seattle, Washington, United States of America, 5 Department of Medicine, University of Washington, Seattle, Washington, United States of America

* tgujral@fredhutch.org (TSG); ageballe@fredhutch.org (APG)

**Data Availability Statement:** All relevant data are within the manuscript and its Supporting information files except for sequence data which were deposited in NCBI BioProject PRJNA884721.

## Abstract

Kaposi's sarcoma-associated herpesvirus (KSHV) causes several human diseases including Kaposi's sarcoma (KS), a leading cause of cancer in Africa and in patients with AIDS. KS tumor cells harbor KSHV predominantly in a latent form, while typically <5% contain lytic replicating virus. Because both latent and lytic stages likely contribute to cancer initiation and progression, continued dissection of host regulators of this biological switch will provide insights into fundamental pathways controlling the KSHV life cycle and related disease pathogenesis. Several cellular protein kinases have been reported to promote or restrict KSHV reactivation, but our knowledge of these signaling mediators and pathways is incomplete. We employed a polypharmacology-based kinome screen to identify specific kinases that regulate KSHV reactivation. Those identified by the screen and validated by knockdown experiments included several kinases that enhance lytic reactivation: ERBB2 (HER2 or *neu*), ERBB3 (HER3), ERBB4 (HER4), MKNK2 (MNK2), ITK, TEC, and DSTYK (RIPK5). Conversely, ERBB1 (EGFR1 or HER1), MKNK1 (MNK1) and FRK (PTK5) were found to promote the maintenance of latency. Mechanistic characterization of ERBB2 pro-lytic functions revealed a signaling connection between ERBB2 and the activation of CREB1, a transcription factor that drives KSHV lytic gene expression. These studies provided a proof-of-principle application of a polypharmacology-based kinome screen for the study of KSHV reactivation and enabled the discovery of both kinase inhibitors and specific kinases that regulate the KSHV latent-to-lytic replication switch.

## Author summary

Kaposi's sarcoma-associated herpesvirus (KSHV) causes Kaposi's sarcoma, a cancer particularly prevalent in Africa. In cancer cells, the virus persists in a quiescent form called latency, in which only a few viral genes are made. Periodically, the virus switches into an active replicative cycle in which most of the viral genes are made and new virus is

**Funding:** This work was supported by grants from the Fred Hutchinson Cancer Center Pathogen-associated Malignancies Integrated Research Center, from the Fred Hutchinson Cancer Center Human Biology Division, from the National Institutes of Health grants RO1AI45945 (to A.P.G.), R01CA189986, R01CA217788, and R21CA240479 (to M.L.), and P30 CA015704; and from the National Science Foundation grant 2047289 (to T. S.G.). The content is solely our responsibility and does not necessarily represent the official views of the National Institutes of Health. The funders had no role in study design, data collection and analysis, decision to publish, or preparation of the manuscript.

**Competing interests:** The authors have declared that no competing interests exist.

produced. What controls the switch from latency to active replication is not well understood, but cellular kinases, enzymes that control many cellular processes, have been implicated. Using a cell culture model of KSHV reactivation along with an innovative screening method that probes the effects of many cellular kinases simultaneously, we identified drugs that significantly limit KSHV reactivation, as well as specific kinases that either enhance or restrict the KSHV replicative cycle. Among these were the ERBB kinases, which are known to regulate growth of cancer cells. Understanding how these and other kinases contribute to the switch leading to production of more infectious virus helps us understand the mediators and mechanisms of KSHV diseases. Additionally, because kinase inhibitors are proving to be effective for treating other diseases including some cancers, identifying ones that restrict KSHV replicative cycle may lead to new approaches for treating KSHV-related diseases.

## Introduction

Kaposi's sarcoma-associated herpesvirus (KSHV) is the etiologic agent of Kaposi's sarcoma (KS), a leading cause of cancer in Africa and a substantial health concern for AIDS patients worldwide [1–3]. KSHV causes three other less prevalent diseases: primary effusion lymphoma in B cells, multicentric Castleman disease, and a KSHV inflammatory cytokine syndrome. The main proliferating tumor cell of KS is the spindle cell. In KS spindle cells, the virus exists predominantly in a latent state in which only a few of its ~90 genes are expressed. Approximately 5% of spindle cells express markers of the lytic replicative cycle [4–7] representing a relatively infrequent switch from latency to lytic replication in tumors. While considerable effort has been devoted to characterizing cellular and viral factors that support the maintenance of latency or the induction of lytic replication, our knowledge of the complex signaling involved in this replicative switch remains incomplete. Because many latent and lytic KSHV genes have oncogenic properties and are involved in disease progression [4,8–13], understanding the regulators of this replicative switch is of fundamental importance for understanding KSHV disease pathogenesis and has potential relevance for new therapeutic interventions.

The human kinome comprises 518 protein kinases known to regulate myriad host and viral processes, including KSHV latency and the switch to lytic replication [14–16]. Due to the essential regulatory roles of kinases, dysregulation of their catalytic activity causes many types of cancers and other diseases. Viruses also usurp cellular kinases or encode their own kinases to modulate the signaling of the host cell to promote specific virus lifecycle stages or replicative functions. For KSHV, both the virus-encoded kinase, ORF36 [17], and cellular kinases are necessary for lytic replication [18–22]. Prior reports identified several kinases with roles in KSHV reactivation using various screening approaches, including a kinase cDNA overexpression screen [23], phospho-site antibody microarray [18], and proteome analysis [24] following KSHV primary infection or after induction of lytic replication. From these and other studies, fewer than a dozen kinases have been validated as aiding latency or facilitating reactivation [18,24,25]. By completing a more comprehensive investigation of protein kinase regulators of KSHV reactivation, we might identify FDA-approved kinase inhibitors (KIs) that could be repurposed with the aim of reducing KSHV reservoirs and/or treating KSHV-associated cancers and lymphoproliferative diseases.

Recently, kinase-centric polypharmacology-based screens have been developed to identify both KIs and their targeted kinases that regulate cell death, cancer cell migration, and other cancer cell phenotypes [26–28]. These screens have also been used to evaluate kinase roles in

*Plasmodium* infected cells and during virus-induced cytokine production [29,30]. This innovative approach employs broadly acting KIs as tools that exploit built-in redundancy from their shared kinase targets, when used in Kinase Inhibitor Regularization (KiR) analyses [26,27,31]. Specifically, the polypharmacology-based kinome screen and KiR platform use a small set of computationally-derived KIs to restrict the catalytic activity of multiple endogenously expressed kinases. Data derived from testing the phenotypic effects of these inhibitors, coupled with known drug specificities and potencies for each kinase target, allows for a network-based, machine-learning analysis that initially predicts the impact of untested KIs. Refinements by iterative screening of additional KIs curates single kinases predicted to have significant regulatory potential for the system evaluated. This method is attractive compared to alternative kinome screening methods due to the high-throughput nature and built-in redundancy for enhanced accuracy. As well, this screen generates two outputs, predicted kinases and KIs, and some of these drugs are FDA-approved.

Herein, we describe the adaption of this kinome screening approach to study KSHV reactivation in an epithelial cell system commonly used to study KSHV reactivation. From this approach, we discovered two drugs, lestaurtinib and K252a, as potent inhibitors of induced lytic replication and eight kinases not previously associated with reactivation. Among these predicted kinases, MKNK2 and ERBB4 had the greatest pro-lytic phenotypes. MKNK2 (MNK2) and the closely related MKNK1 (MNK1) are both mitogen-activated kinases and are the only kinases known to phosphorylate eIF4E [32]. The epidermal growth factor receptor kinase ERBB4 has three other family members, all of which regulate numerous cell-signaling pathways including transcription, translation, cell survival, and cell growth [33,34]. Evaluation of the other members of these kinase families revealed that ERBB2 (HER2 or *neu*) and ERBB3 are pro-lytic factors, like MKNK2 and ERBB4, while MKNK1 and ERBB1 (EGFR) have opposite, pro-latent effects that restrict lytic replication. Characterization of ERBB2 signaling during KSHV reactivation revealed a new connection between ERBB2 and activation of CREB1, a transcription factor known to activate KSHV lytic gene expression [18]. Based on our findings, we propose a model in which ERBB1:ERBB2 heterodimers may exist and signal to promote latency. Next, induction of lytic replication turns on ERBB3 expression, providing a higher affinity-binding partner for ERBB2 that can steal ERBB2 away from ERBB1. Due to similar lytic-promoting phenotypes of ERBB2 and ERBB3 and to a lesser degree ERBB4, our working model suggests that newly formed ERBB complexes signal through CREB1 and STAT3 to promote lytic replication. These findings provide initial insights into how the KSHV latent-to-lytic replication switch can be regulated by interfering with ERBB signaling and suggest the potential utility of manipulating signaling from these plasma membrane receptors as a new therapeutic approach.

## Results

### Generation of a recombinant KSHV with infection and lytic replication indicators

To enable precise measurement of the initial transition from KSHV latent-to-lytic replication, we constructed a new recombinant virus called lytic replication indicator KSHV (KSHV$^{\text{LRI}}$). This virus is derived from the KSHV bacterial artificial chromosome 16 (BAC16) that constitutively expresses GFP, enabling identification of infected cells. For detecting lytic replication, we introduced an expression cassette containing the KSHV polyadenylated long non-coding RNA promoter (PrPAN) to drive the expression of a streptavidin-binding peptide fused to a truncated low-affinity nerve growth factor receptor (SBP-ΔLNGFR) and nuclear localized mCherry (mCherry-NLS) (Fig 1A). The coding regions for these two genes are separated by a

**A.**

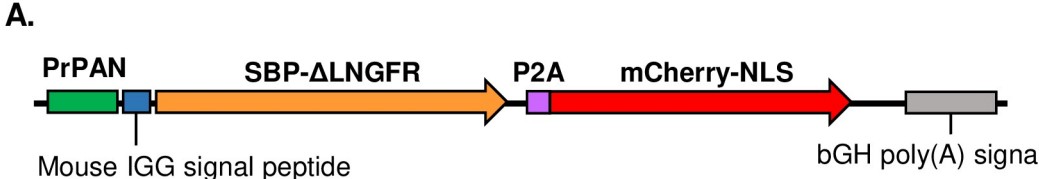

**B.**

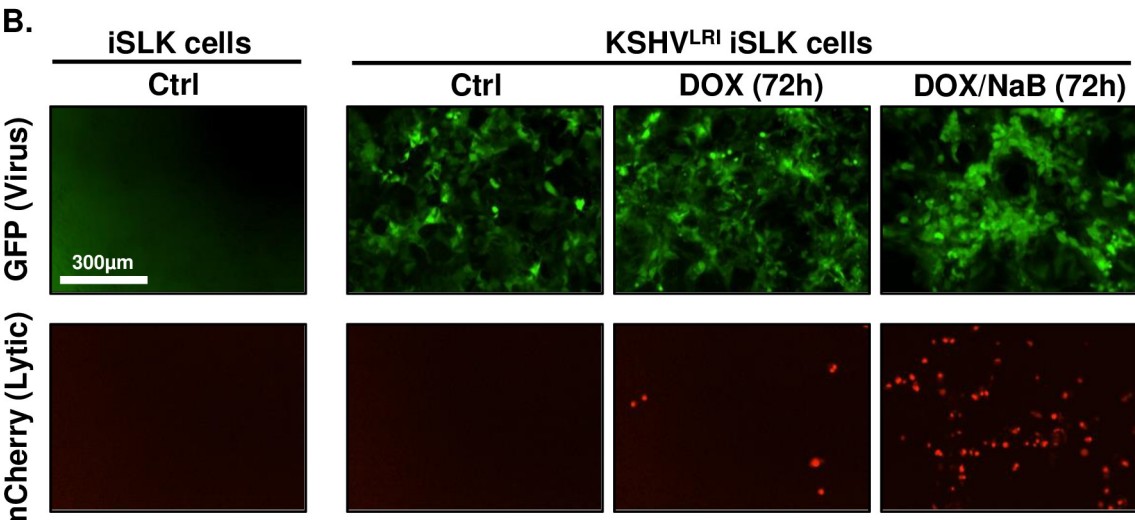

**Fig 1. Characterization of KSHV dual lytic replication indicator virus (KSHV$^{LRI}$) reactivation in iSLK cells. (A)** Diagram of DNA cassette inserted into KSHV BAC16 genome to generate the KSHV$^{LRI}$ recombinant. The KSHV *PAN* promoter (PrPAN) drives expression of the streptavidin-binding peptide fused to a truncated low-affinity nerve growth factor receptor (SBP-ΔLNGFR), a P2A self-cleaving peptide sequence, and then mCherry containing a nuclear localizing signal (mCherry-NLS). **(B)** Fluorescence images of uninfected iSLK and KSHV$^{LRI}$ infected cells collected using an Incucyte Imaging System. Infected cells express the virus encoded *GFP* under control of the cellular EF-1α promoter (top row). mCherry-nls (bottom row) detected after inducing lytic replication with 1μg/ml DOX (middle column) or DOX plus 1mM NaB (right column) for 72h.

2A self-cleaving peptide (P2A) sequence. The SBP-ΔLNGFR was designed for selection by streptavidin binding of cells undergoing lytic replication. The NLS on mCherry enables accurate, high-throughput quantification of individual cells containing lytic replicating virus by fluorescence imaging. We introduced KSHV$^{LRI}$ into iSLK cells, a human renal carcinoma epithelial cell line that encodes doxycycline (DOX)-inducible KSHV replication and transcription activator (RTA). As expected, the KSHV$^{LRI}$ infected cells expressed GFP (Fig 1B) and LANA (S1A and S1B Fig), and addition of lytic inducing agents, DOX or DOX plus sodium butyrate (NaB), resulted in mCherry-NLS (Fig 1B) and SBP-ΔLNGFR (S1C and S1D Fig) expression. Importantly, the engineered fluorescent constructs allowed us to monitor total KHSV$^{LRI}$ infected cells by GFP area and virus reactivation by mCherry-NLS positive iSLK cells using high-throughput, quantitative fluorescence imaging.

## Polypharmacology-based kinome screen for enhancers or repressors of KSHV reactivation

To identify kinases important for KSHV reactivation, we employed a polypharmacology-based kinome screen using the Kinase inhibitor Regularization (KiR) pipeline as described in our previously published manuscript and supporting information [31]. The KiR approach is based

on the fact that all KIs inhibit multiple kinases but the breadth and potency of each KI for the targeted kinases provides a unique profile. In this approach, an "optimal set" of KIs is computationally derived from a matrix of inhibition profiles of 427 compounds against 371 kinases [31]. Unsupervised hierarchical clustering is applied to split the drugs into distinct clusters based on their inhibitory profile. From each cluster, a final collection of drugs is chosen to ensure coverage of many potential kinases, with built-in redundancy since each kinase's residual activity is reduced at a minimum by 30 percent by at least two drugs from this selected set [26]. This results in the collection of initial inhibitors used in the screen.

The lytic replication indicator KSHV enabled the adaptation of the KiR screen to predict KIs and kinases that either promote or restrict KSHV reactivation. First, we incubated KSHV^LRI latently infected iSLK cells with the vehicle (DMSO) control or each of the 29 computationally-selected KIs (at four concentrations (31nM, 125nM, 500nM, 2μM) and concurrently induced lytic replication by adding DOX plus NaB to the cell media (Fig 2A). At 72h post-treatment, we quantified KSHV reactivation by counting mCherry-NLS positive cells per image and cell survival by measuring GFP area per image using the Incucyte Imaging System (S1 Table). A representative dose-response curve for two KIs (cmpd #1: CP-547632, CAS no. 252003-65-9; cmpd #2: NCGC00263020-01, CAS no. 1221153-14-5) that either enhanced or restricted KSHV reactivation at 500nM concentration is shown in Fig 2B. Several KIs were removed from further analyses due to reduced cell confluence by > 30% as compared to the DMSO DOX+NaB control, suggesting possible cell toxicity, or to inconsistent dose responses. Data from the remaining 19 KIs at 500nM with DOX plus NaB resulted in a range of 6–30% total cells reactivated and provided the initial "training set" for machine learning-based analyses [35]. For the DMSO control, 20% total cells were reactivated and set to 100 (Fig 2C). While there were changes in the induction of lytic replication following treatment with only the KIs (without DOX or NaB), the range of lytic replication (0–0.06% total cells) was too small for the KiR analysis to generate KI predictions (S2 Fig). These data suggest that broad kinase inhibition alone cannot efficiently activate the switch from latency to lytic replication in this system, but following RTA expression and release of some epigenetic constraints by NaB, kinases do measurably regulate KSHV reactivation.

For the lytic inducing condition (DOX + NaB), which had a greater dynamic range for measuring KSHV reactivation, we produced an initial KiR model from the 19 KI "training set" phenotype dataset and a separate dataset of >400 KI's effects on 298 recombinant human protein kinases [35]. Leave-one-out-cross validation (LOOCV) was used to evaluate the accuracy of the model. We then tested an additional 12 KIs predicted by the KiR model to impact KSHV reactivation (Fig 2D, S2 Table). The resulting responses for 7 of these KIs (S5 Table) were iteratively included in the training set to improve the accuracy of the model. Based on the final KiR model, the 27 KIs included in the analysis targeted 354 of the 371 profiled kinases (~95% of the measured kinome) and the correlation between predicted and observed responses was >0.8. The final list of 427 predicted KIs is shown in Fig 2E and S3 Table. Of note, two broadly acting KIs, lestaurtinib and K252a, were predicted to be regulators of reactivation based on the initial KiR model. Following the testing of these drugs (Fig 2D) and the inclusion of the KSHV reactivation data into the final model, these two drugs remained at the top of the list of KIs predicted to regulate KSHV reactivation (S3 Table).

## Kinases validated as cellular regulators of KSHV reactivation in epithelial cells

Based on the KiR model, we selected the top 13 "most informative" kinases with the highest positive coefficient (S4 Table). PKG2 was excluded due to lack of expression data in latent and

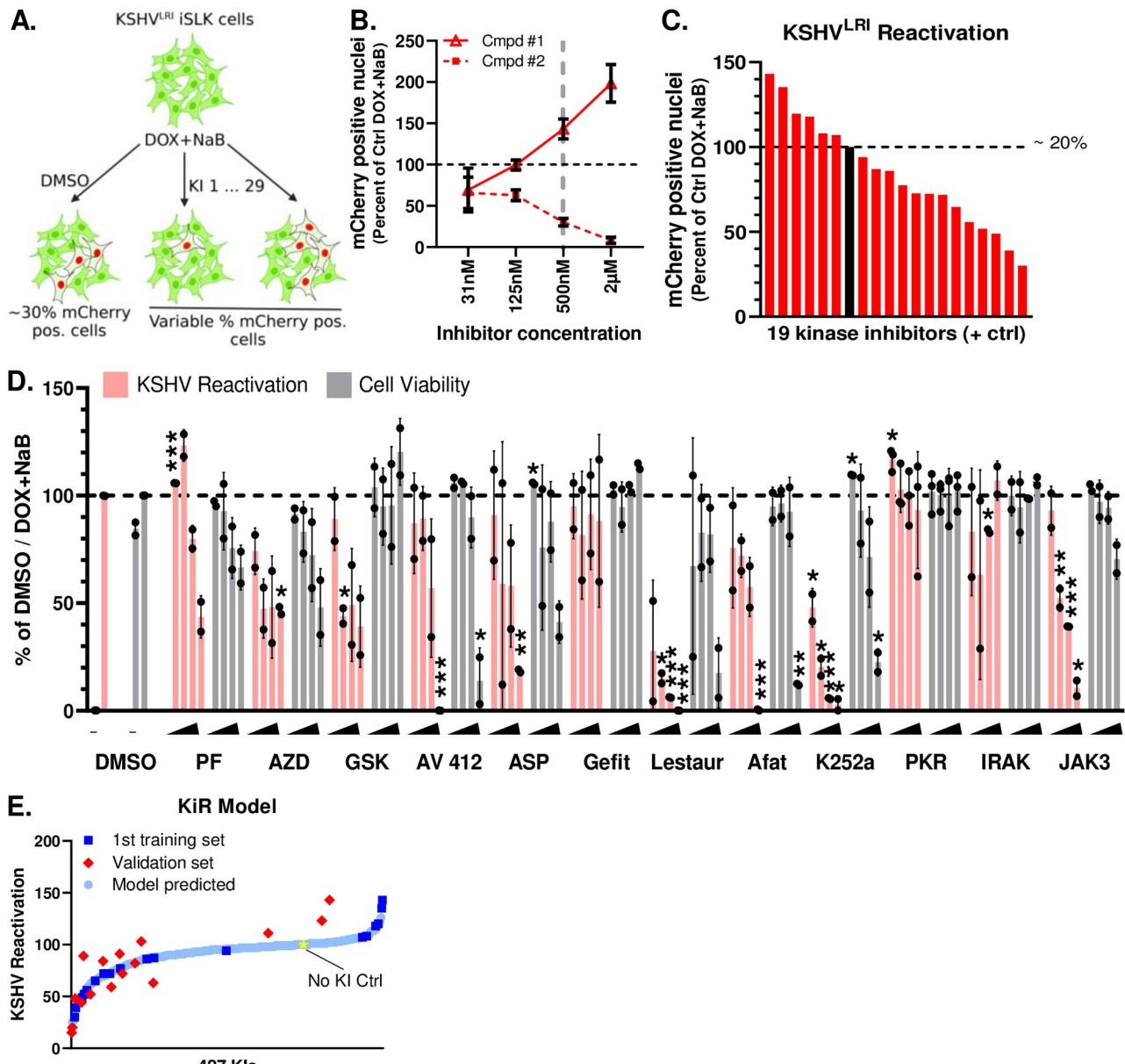

**Fig 2. Polypharmacology-based kinome screen to identify kinases important for restriction or enhancement of KSHV lytic replication. (A)** 29 selected kinase inhibitors (KIs) were tested at four concentrations (0.031, 0.125, 0.5, and 2uM) in KSHV$^{LRI}$ latently infected iSLK cells, in combination with KSHV lytic replication inducing agents. KSHV reactivation was measured by Incucyte Imaging System quantification of mCherry-NLS expression at 72h following treatment and provided the phenotypic data for the machine-learning analyses and prediction of kinases important for KSHV reactivation. **(B)** Data from testing two of these proprietary compounds illustrate dose-dependent increased or restricted KSHV reactivation. **(C)** KSHV$^{LRI}$ reactivation phenotypes were obtained from 19 of the 29 selected KIs that had minimal changes ($< 30\%$) to cell confluences as compared to the DMSO DOX+NaB control as a measurement of cell viability and that demonstrated consistent dose responses curves. Reactivation data for DOX plus NaB treatment alone (black bar and dotted black line set to 100, represents ~20% total cells) or combined with a KI (red bars, 0.5 μM) were calculated as a percent of DOX plus NaB control. **(D)** Twelve additional KIs predicted from the initial model were tested for effects on KSHV reactivation (pink bars) and cell viability (grey bars). The iSLK KSHV$^{LRI}$ cells were untreated ("-" DMSO only) or treated with DOX plus NaB alone (DMSO) or in combination with KIs at 125nM, 500nM, 2μM, or 8μM. Reactivation (pink bars) for each KI condition was measured by mCherry expression and calculated as a percent of DOX plus NaB with DMSO control. Cell viability (grey bars) was determined by cell confluence as a percent of DOX plus NaB with DMSO control. PF: PF-477736; AZD: AZD3463; GSK: GSK-650394; ASP: ASP-3026; Gefit: Gefitinib; Lestaur: Lestaurtinib; Afat: Afatinib; PKR: PKR Inhibitor; IRAK: IRAK1/4; JAK3: JAK3 Inhibitor VI. Unpaired t tests were performed in Excel for each kinase inhibitor condition compared to the DMSO control and p-value symbols denoted $* \leq 0.05$, $** \leq 0.01$, $*** \leq 0.001$. **(E)** The KiR model was developed by incorporating data on KI s as the training set (C). This data was used to predict the response of potential KIs. A subset of these predictions was then tested as a validation set (D), and the results were used to refine the model's predictions. The No KI Ctrl (yellow star) represents the DOX plus NaB with DMSO control condition. Diagrams in Fig 2A were generated using BioRender.

lytic induced iSLK cells. Our KiR training set and final predicted KIs includes coverage of the 13 predicted kinases (S3 and S5 Tables). We evaluated these predicted kinases using pooled siRNAs to target each, which generally achieved efficient knockdown (S3 and S4 Figs). Individual depletion of six of these kinases significantly altered KSHV reactivation under lytic inducing conditions without having significant effects on cell viability (Fig 3). Knocking down ERBB4 (HER4), MKNK2 (MNK2), and DSTYK (RIPK5) as well as two TEC family kinases, ITK and TEC, reduced reactivation, indicating that these kinases are pro-lytic factors. In contrast, knocking down FRK (RAK or PTK5) caused a slight but statistically significant increase in reactivation, suggesting that it may be a pro-latency factor.

Our finding that knocking down ITK reduced reactivation (Fig 3) was surprising since we did not detect ITK expression by RT-qPCR in our latently infected cells. Consistent with our expression results, RNA-seq data from KSHV BAC16 latently infected iSLK cells showed very low ITK expression ([36], S6 Table). However, that study also revealed a >5-fold increase in ITK expression at 48h post lytic induction. The observation that knocking down another TEC family kinase (TEC) also restricted reactivation supports the conclusion that TEC kinase family signaling likely contributes to KSHV reactivation.

Because knocking down MKNK2 had the strongest inhibitory effect on KSHV reactivation and has another family member with shared substrates and overlapping signaling pathways, we tested the impact of knocking down this related kinase using specific siRNAs (S3B Fig). Intriguingly, MKNK1 knockdown had the opposite effect of increased reactivation, indicating a pro-latent effect of MKNK1, as compared to the pro-lytic activity of MKNK2 (Fig 3B).

Together, these data identify MKNK2, ERBB4, ITK, TEC, and DSTYK as kinases that contribute to KSHV reactivation while MKNK1 and FRK restrict it. One prior study reported that a KI that moderately restricts MKNK1 and MKNK2 catalytic activity, reduced lytic replication [37] but otherwise, to the best of our knowledge, all the kinases we validated represent uncharacterized regulators of KSHV.

## ERBBs differentially regulate KSHV latent-to-lytic replicative switch

The second strongest phenotype was observed for ERBB4 knockdown which significantly reduced KSHV reactivation (Fig 3). Similar to the MKNKs, the ERBB4 protein shares overlapping signaling pathways and substrates with three other family members: ERBB1 (EGFR, HER1), ERBB2 (HER2, *neu*), and catalytically inert ERBB3 (HER3). Intriguingly, published gene expression data in KSHV infected iSLK cells demonstrated that ERBB1 and ERBB2 were similarly expressed under latent and lytic induced conditions while ERBB3 and ERBB4 were detectable only after treating with lytic inducing agents (S6 Table, [36]). To evaluate ERBB protein abundance under both latent and lytic induced conditions, we performed immunoblot assays with our cells and compared these with two breast cancer cell lines in which the ERBBs have been well-studied [38]. ERBB1 and ERBB2 total protein remained relatively similar between latent and lytic induced conditions and ERBB2 levels were much less than in BT474 cells, which overexpress ERBB2 (Fig 4A). Additionally, the negligible levels of ERBB3 and ERBB4 in latent cells were moderately increased for ERBB3 and slightly increased for ERBB4 under lytic inducing conditions. Since ERBB2 is the preferential binding partner of all other ERBBs [39], we assayed for ERBB2 phosphorylation (P-Tyr$^{1221/1222/1248}$) as an indicator of ERBB2 active dimer or oligomerization [40,41]. Immunoblot of ERBB2 Tyr$^{1248}$ phosphorylation (Fig 4B) or reverse phase protein assay of Tyr$^{1221/1222}$ phosphorylation (Fig 4C) similarly demonstrated active, phosphorylated ERBB2 under both latent and lytic inducing conditions.

We next assessed the role of each ERBB kinase on reactivation by depleting one of the four ERBB kinases (S4 Fig) and performing our mCherry KSHV reactivation assay. Strikingly,

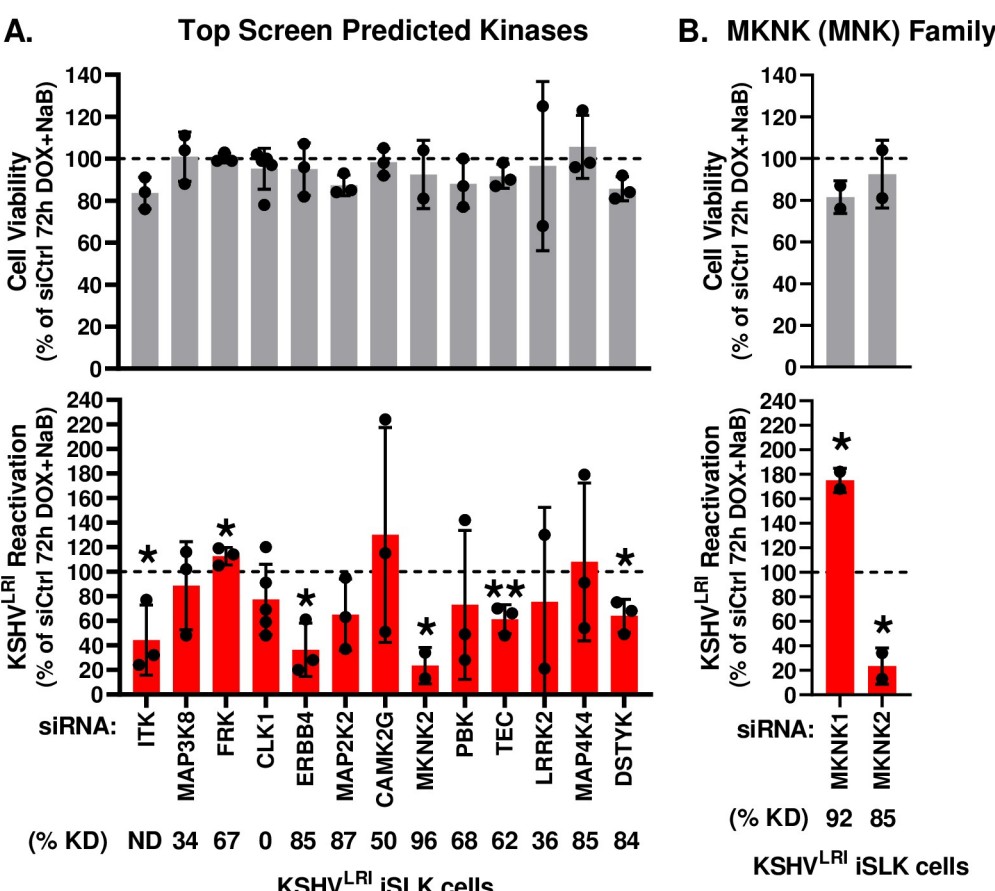

**Fig 3. Validation of predicted kinases and MKNK1 to regulate KSHV reactivation.** For the **(A)** predicted kinases and **(B)** MKNK family, cell viability (grey bars) and KSHV reactivation (red bars) were measured for siRNA treated cells at 6-days post siRNA transfection and 3-days post DOX and NaB treatment. Control siRNA transfected cells treated with DOX plus NaB (dotted black line) was set to 100 and data for each kinase knockdown condition was calculated as a percent of the control. Kinase knockdown efficiencies (% KD) at 3-days following siRNA transfection were determined before addition of lytic inducing drugs by harvesting total RNA and performing RT-qPCR as graphed in S3 Fig and for ERBB1-4 in S4 Fig. For each knockdown, the efficiencies were calculated by averaging siKinase / siCtrl and listed below the corresponding kinase target as % KD. For ITK, the relative mRNA levels were not detectable (ND) in these cells using RT-qPCR. Unpaired t tests were performed in Excel for each kinase knockdown condition compared to siCtrl. P-value * < 0.05 and ** < 0.01.

ERBB2 and ERBB3 showed pro-lytic functions (Fig 4D) similar to ERBB4. ERBB1 knockdown had a similar trend but was not statistically significant. To investigate the essentiality of ERBB2 and ERBB1 catalytic activity during KSHV reactivation, we tested the effect of the KI tucatinib. This drug inhibits ERBB2 specifically at low concentrations, and also inhibits ERBB1 at higher concentrations [42]. Consistent with ERBB2 playing a role in reactivation, low concentrations of the drug (31nM) modestly reduced KSHV reactivation (Fig 4E). At 125nM, levels of reactivation returned to control levels, possibly due to tucatinib also inhibiting ERBB1, knockdown of which had little effect on KSHV reactivation in Fig 4D.

Since three of the ERBB kinases shared pro-lytic properties, we prioritized further study of these kinases in our KSHV infection system. First, we quantified the impact of depleting each ERBB on the transcription of the lytic *RTA* (an early gene and/or DOX-induced), *PrPAN-mCherry* (an early gene), *ORF10* (a late gene), and *K8.1* (a late gene) in latently infected cells

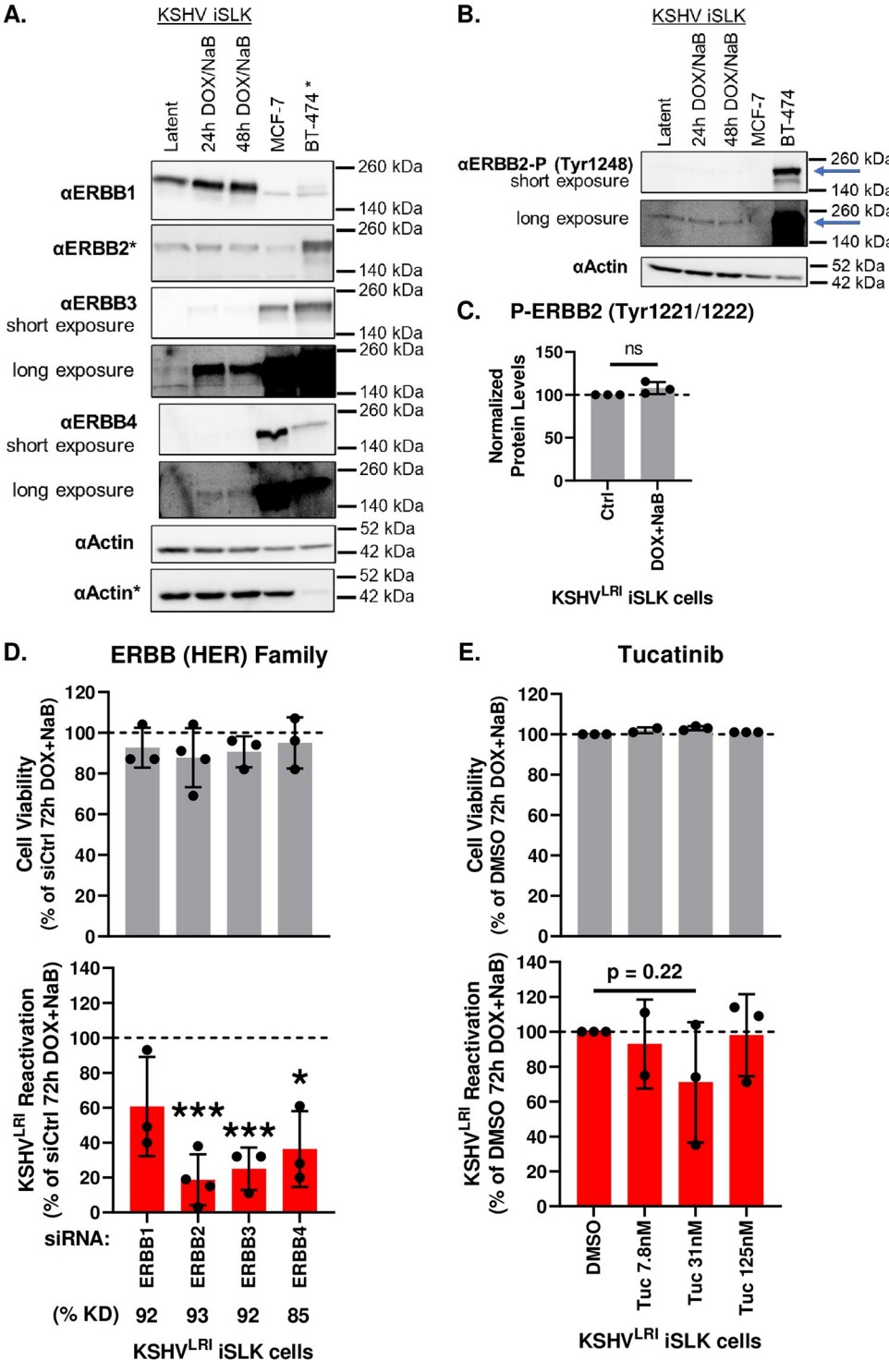

**Fig 4. Effects of EBBB family kinases on KSHV reactivation.** (**A**) Expression of the ERBB family members in latent and lytic induced KSHV infected iSLK cells and two well-characterized control cells lines (MCF-7 and BT474) was measured by immunoblot assays. * indicates that 10% as much BT474 lysate was loaded on the gel probed for ERBB2 and the lower actin blot. Phosphorylation of EBBB2 was measured (**B**) using antibodies to Tyr[1248] by immunoblot (blue arrow) and (**C**) with antibodies to ERBB2 Tyr[1221/1222] by reverse-phase protein arrays. (**D**) Cell viability (grey bars) and KSHV reactivation (red bars) were measured at 6-days post-transfection with siRNAs targeting each ERBB and 3-days post DOX and NaB treatment. Control siRNA transfected cells treated with DOX plus NaB (dotted black line) was set to 100 and data for each kinase knockdown condition was calculated as a percent of the control. Kinase

knockdown efficiencies (% KD) at 3-days following siRNA transfection were determined before addition of lytic inducing drugs by harvesting total RNA and performing RT-qPCR as graphed in S4 Fig. For each knockdown, the efficiencies were calculated by averaging siKinase / siCtrl and listed below the corresponding kinase target as % KD. **(E)** Cell viability (grey bars) and KSHV reactivation (red bars) were measured at 3 days after lytic induction, in the absence (DMSO) or presence of the indicate concentrations of tucatinib. Unpaired t tests were performed in Excel for each kinase knockdown of KI treatment condition compared to siCtrl or DMSO, respecitvely. P-value * < 0.05 and *** < 0.001.

and upon incubation with lytic inducing agents. Under latent conditions, knockdown of individual ERBBs had no effect on the low levels of *RTA* (Fig 5A). Expression of the other viral genes was not detected under latent conditions. For lytic inducing conditions, knockdown of each of the ERBB did not affect *RTA* expression (Fig 5B). Therefore, under both conditions, ERBB depletions do not significantly change *RTA* levels. To our surprise, knockdown of

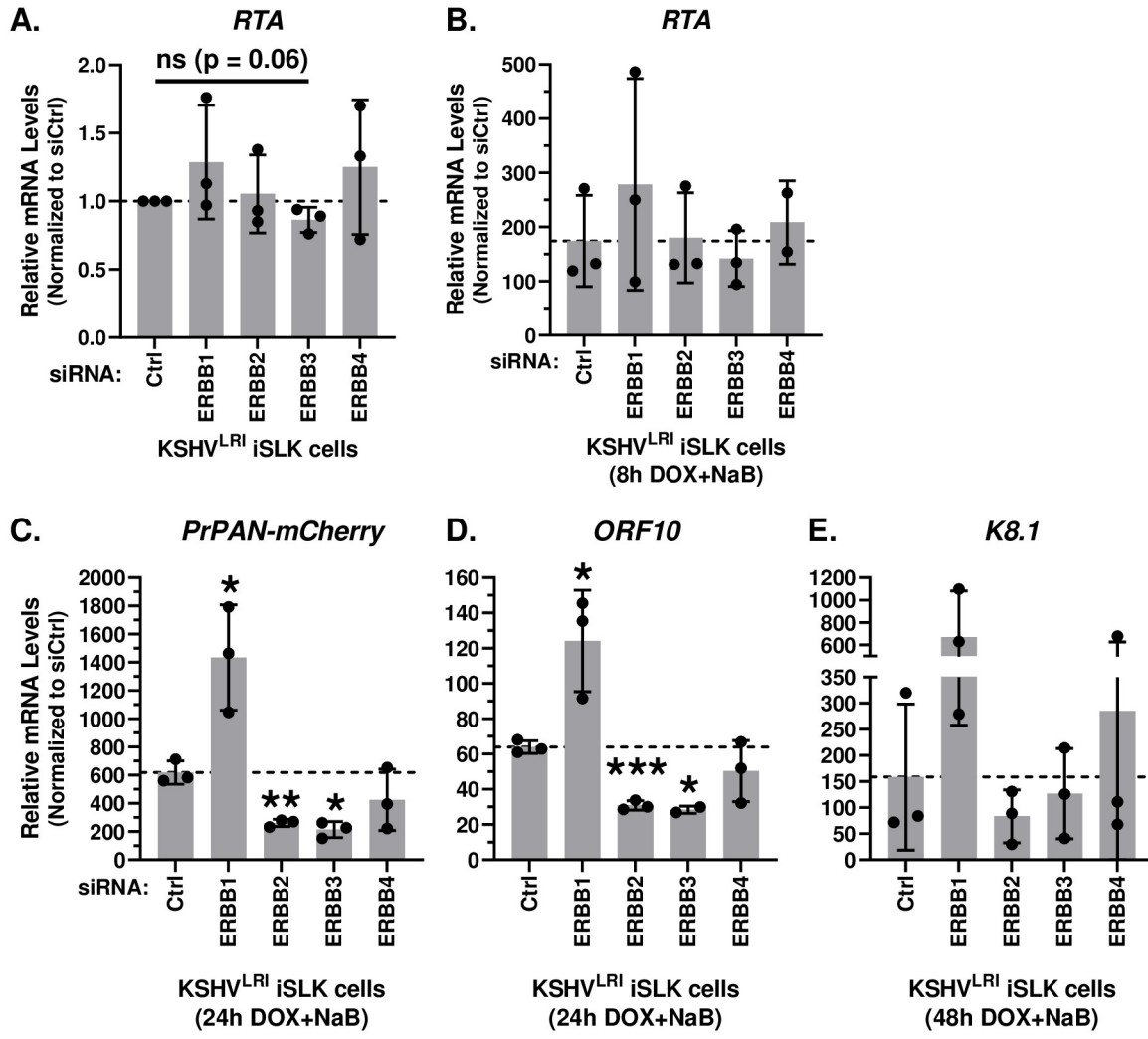

**Fig 5. Effects of ERBB family member depletion on lytic gene transcript accumulation.** KSHV[LRI] latently infected iSLK cells were transfected with siRNA control or siRNAs targeting individual ERBB family kinases and then 3-days later treated with DOX plus NaB for 24h or 48h. Transcript levels for **(A, B)** *RTA*, **(C)** *PrPAN-mCherry* **(D)** *ORF10* and **(E)** *K8.1* under (A) latent and (B-E) lytic inducing conditions were quantified using RT-qPCR with target specific primers. Unpaired for (A) and paired for (B-E) t tests were performed in Excel for each kinase knockdown condition compared to siCtrl.

ERBB1 significantly increased lytic *PrPAN-mCherry* and *ORF10* transcript abundance and had a similar trend for late lytic *K8.1* expression (Fig 5C–5E). These data do not correlate with the KSHV reactivation data in Fig 4D and suggest that ERBB1 may signal through at least two distinct pathways to regulate KSHV reactivation. ERBB2 and ERBB3 knockdown significantly reduced both *PrPAN-mCherry* and *ORF10* expression in lytic induced cells (Fig 5C and 5D), consistent with the mCherry phenotype (Fig 4D). Expression of *K8.1* was restricted by either ERBB2 or ERBB3 knockdown, but these changes were not statistically significant (Fig 5E). These data support roles for ERBB1 to limit KSHV lytic replication or perform pro-latent functions, while ERBB2 may have differing roles in reactivation depending on the conditions. ERBB3 and to a lesser degree ERBB4 are primarily expressed following lytic induction, at which time they enhance KSHV reactivation.

## ERBB2 phosphorylation of ERBB1 signaling is disrupted during KSHV latent-to-lytic replication switch

Because ERBB2 is the preferential binding partner of all other ERBBs [39] and ERBB3 cannot signal alone [43], we focused on ERBB2 signaling during KSHV reactivation. The ERBB signaling cascades are complex [33,34]. Therefore, we measured phosphorylation of a panel of substrates (Fig 6A) using a high-throughput, reverse-phase protein array approach under both latent and induced lytic replication conditions in control and ERBB2 depleted cells. Depletion of ERBB2 in latently infected cells significantly attenuated the activation of ERBB1, as measured by phosphorylation at Tyr[1173] (Fig 6B). This result suggests that in latent cells, ERBB1:ERBB2 heterodimers form and ERBB2 transphosphroylates ERBB1 to activate downstream signaling. Treatment with lytic inducing agents (in control siRNA cells) attenuated ERBB1 phosphorylation to a similar extent as in latently infected cells with ERBB2 depletion. ERBB2 knockdown in cells treated with lytic inducing agents did not further reduce ERBB1 phosphorylation. These results suggest that the ERBB1:ERBB2 heterodimer is disrupted during

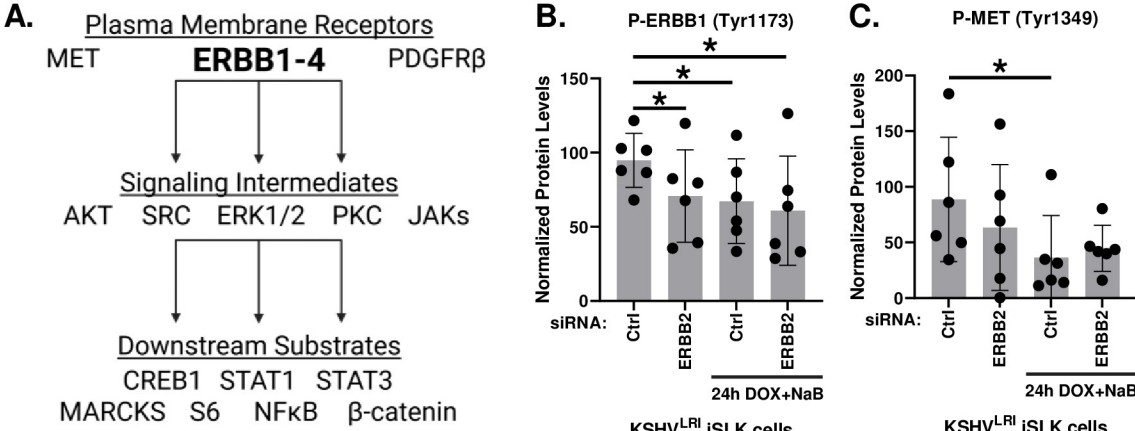

**Fig 6. ERBB2 effects on ERBB1 and MET phosphorylation. (A)** Diagram of selected proteins tested to elucidate ERBB2 signaling. **(B)** KSHV[LRI] latently infected iSLK cells were transfected with siRNA control or siRNAs targeting ERBB2 and then 3-days later untreated or treated with DOX plus NaB for 24h. Cells were harvested, and protein lysates were analyzed using a reverse-phase protein array (RPPA) for ERBB1 phosphorylation at Tyr[1173] or **(C)** for MET phosphorylation at Tyr[1349]. Relative phospho-protein levels were normalized to untransfected control cells (Ctrl) by setting this to 100 on the y-axis. Paired t tests were performed in Excel for each kinase knockdown condition compared to siCtrl or siCtrl with 24h DOX+NaB. P-value * $\leq$ 0.05. Diagrams in Fig 6A were generated using BioRender.

induction of lytic replication and are consistent with our finding that ERBB1 is a pro-latent factor that restricts lytic gene expression (Fig 5).

In our panel of substrates, we also evaluated the phosphorylation of proteins involved in parallel signaling or crosstalk with EBBB2, as well as downstream signaling intermediates, and substrates of the selected signaling intermediates (Fig 6A). For proteins involved in crosstalk signaling, we measured the phosphorylation of plasma membrane receptors mesenchymal epithelial transition (MET) [44–46] and platelet derived growth factor receptor beta (PDGFRβ) [47] during lytic reactivation in control and ERBB2 knockdown cells. Similar to ERBB1, we found that MET phosphorylation decreased during lytic induction as compared to latent infection (Fig 6C), however, PDGFR-β was unchanged (S5A Fig). Furthermore, ERBB2 knockdown did not affect MET or PDGFR-β phosphorylation, indicating that neither is regulated by ERBB2 during latency or reactivation.

In testing downstream signaling intermediates, we found that depletion of ERBB2 significantly reduced AKT phosphorylation during latency (Fig 7A). Phosphorylation of SRC and ERK1/2 showed a similar reduced trend but was not statistically significant. Strikingly, the phosphorylation of all these kinases was similarly decreased following reactivation. Depleting ERBB2 during reactivation did little to alter these trends, although it did slightly increase ERK1/2 phosphorylation as compared to control lytically induced cells (Fig 7A). No significant trend was observed for pan PKC phosphorylation at PKC βII Ser[660] and homologous residues on the other PKC isoforms (S5B Fig). The reduced phosphorylation of AKT, SRC and ERK1/2 mimicked the decreased ERBB1 phosphorylation phenotypes and suggest that ERBB1:ERBB2 signaling through AKT and to a lessor degree SRC and ERK1/2 is ERBB2 dependent in latent cells and disrupted during early stages of lytic replication.

The janus kinase family (JAKs) are also signaling intermediates downstream of the ERBBs. The JAK3 Inhibitor VI broad KI (Fig 7B, S7 Table), predicted from the initial KiR screen to affect reactivation, was confirmed to regulate lytic reactivation (Fig 2D). To further investigate the role of JAK signaling, we tested another JAK inhibitor, tofacitinib that has greater specificity for the JAK proteins (Fig 7B) and found that it did not affect reactivation (Fig 7C). Since tofacitinib inhibits the catalytic activity of all JAK kinases while the JAK3 Inhibitor VI is more specific for JAK3 and TYK2 JAKs and several other unrelated kinases (Fig 7B), we tested if different JAK family members may have counteracting effects on reactivation as we observed for the MKNK family protein kinases (Fig 4). We used specific siRNAs to deplete JAK1, JAK2, JAK3, or TYK2 and performed assays under two lytic induction conditions. Under full lytic inducing conditions (DOX + NaB), none of these individual depletions impacted reactivation significantly (Fig 7C). We also conducted experiments in uninduced and DOX-only induced cells and observed that depletion of JAK1 increased reactivation 2.4-fold for the DOX condition as compared to the control (S5C Fig), suggesting that JAK1 is a pro-latency factor. Knockdown efficiency and specificity were confirmed for siRNAs targeting each JAK1, JAK2 and TYK2 using RT-qPCR (S6A Fig). JAK3 RNA abundance was below the level of detection for RT-qPCR. To assess the knockdown efficiency of the siRNAs targeting JAK3, we transiently overexpressed JAK3 by transfecting HeLa cells with a JAK3 expression plasmid and then transfected cells with targeting siRNAs followed by immunoblot analysis of JAK3 protein levels. Exogenously expressed JAK3 was almost completely depleted in this system (S6B Fig). These findings illustrate that the JAK1 protein kinase may have pro-latent activity which could be regulated in parallel with AKT, SRC and ERK1/2, by ERBB1:ERBB2 heterodimeric signaling, or by another upstream receptor that is activated in response to RTA expression.

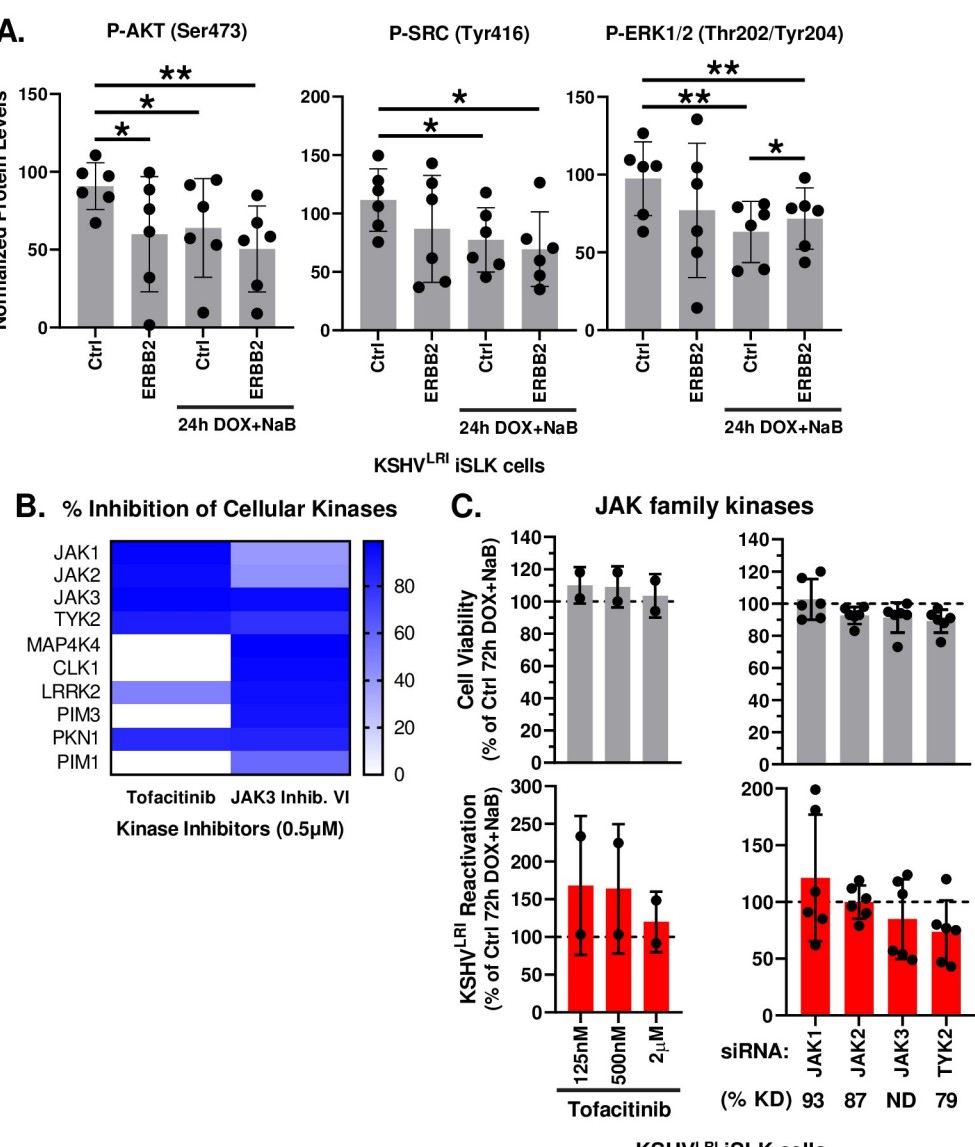

**Fig 7. ERBB2 phosphorylation of intermediate kinases. (A)** KSHV^LRI latently infected iSLK cells were transfected with siRNA control or siRNAs targeting ERBB2 and then 3-days later untreated or treated with DOX plus NaB for 24h. Cells were harvested, and protein lysates were analyzed using a RPPA for the phosphorylation of the indicated signaling intermediates downstream of ERBB2. Relative phospho-protein levels were normalized to untransfected control cells (Ctrl) by setting this to 100 on the y-axis. **(B)** Kinase inhibition profiles for tofacitinib and JAK3 inhibitor VI from the Kinhibition database (https://kinhibition.fredhutch.org/) [26]. JAK3 VI inhibitor restricts JAK3, TYK2 and 15 other kinases. **(C)** Cell viability (grey bars) and KSHV reactivation (red bars) were measured for KSHV^LRI latently infected iSLK cells untreated or tofacitinib treated cells in combination with lytic inducing agents DOX plus NaB for 72h and for cells transfected with siRNAs targeting individual JAK family kinases and 3-days later treated with DOX plus NaB for 72h. Control DMSO or control siRNA transfected cells treated with DOX plus NaB (dotted black lines) were set to 100 and data for each condition was calculated as a percent of the control. Kinase knockdown efficiencies at 3-days following siRNA transfection were determined before addition of lytic inducing drugs and graphed in S6 Fig. For each knockdown, the efficiencies were averaged and listed below the corresponding kinase target as % KD. Paired for (A) or unpaired (C) t tests were performed in Excel for each kinase knockdown condition compared to siCtrl or siCtrl with 24h DOX+NaB or for KI as compared to DMSO control. P-values * $\leq$ 0.05 and ** $\leq$ 0.01.

## ERBB2 and ERBB3 mediate the phosphorylation of CREB1, STAT1, and STAT3 transcription factors during KSHV reactivation

Finally, we tested the role of ERBB2 on the phosphorylation of proteins downstream of the selected signaling intermediates, including myristoylated alanine-rich C-kinase substrate (MARCKS), S6 ribosomal protein, and several transcription factors (Fig 6A). Phosphorylation of the MARKS substrate, which is downstream of PKC or ROCK [48,49], was increased during induction of lytic replication (S5D Fig). ERBB2 knockdown decreased the phosphorylation but not to a statistically significant extent. We detected only small and mostly insignificant effects of lytic induction and ERBB2 knockdown on phosphorylation of S6 and NFκB and on total β-catenin (S5E–S5G Fig). Despite these findings, we did observe a striking phenotype for three other transcription factors. Specifically, the cyclic AMP-responsive element-binding protein 1 (CREB1), signal transducer and activator of transcription 1 (STAT1), and STAT3 all exhibited an increase in phosphorylation under lytic inducing conditions as compared to latency and the effect was attenuated by depletion of ERBB2 in lytically induced cells (Fig 8). Similarly, following lytic induction, ERBB3 knockdown resulted in a slight reduction in CREB1 phosphorylation and decreased STAT1 phosphorylation (S7 Fig). These findings support a signaling function of ERBB2 and ERBB3 to phosphorylate these transcription factors at residues important for transcriptional activity [50–56]. The decrease in CREB1 and STAT1 phosphorylation levels in ERBB4 knockdown cells did not achieve statistical significance (S7 Fig). In KSHV infected cells, phosphorylation of CREB1 Ser$^{133}$ by MSK1/2 was reported to enhance KSHV lytic gene expression [18] and others report that CREB1 promotes replication of other DNA viruses [57–62]. For the STATs, KSHV progeny production is attenuated in STAT3 depleted cells [63] and both STAT1 Tyr$^{701}$ and STAT3 Tyr$^{705}$ are phosphorylated by KSHV early lytic protein vIL-6 which activates cell proliferation, angiogenesis, and tumorigenesis [64–67]. ERBB1 and/or ERBB2 proteins can directly or through indirect signaling pathways regulate CREB1 Ser$^{133}$, STAT1 Tyr$^{701}$ and STAT3 Tyr$^{705}$ phosphorylation in other

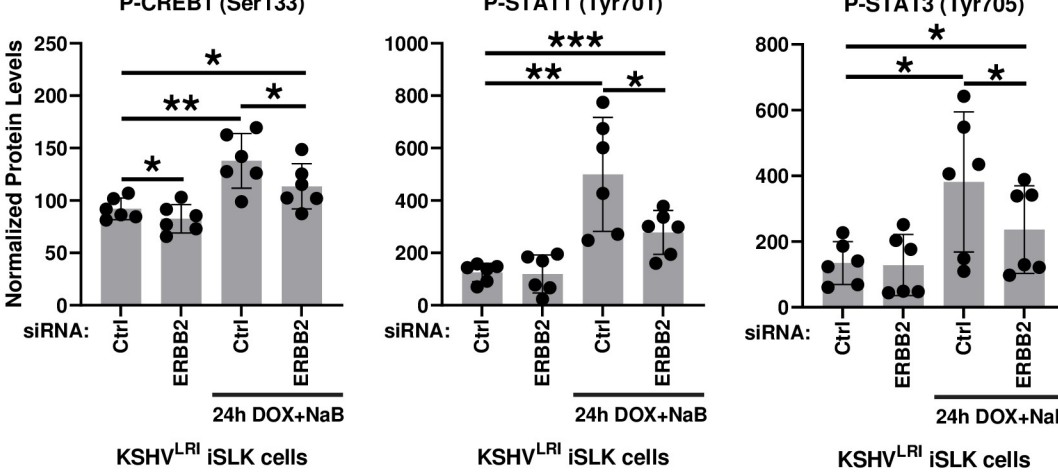

**Fig 8. ERBB2-mediated signaling increases CREB1, STAT1 and STAT3 phosphorylation during lytic replication.** KSHV$^{LRI}$ latently infected iSLK cells were transfected with siRNA control or siRNAs targeting ERBB2 and then 3-days later untreated or treated with DOX plus NaB for 24h. Cells were harvested, and protein lysates were analyzed using a RPPA for phosphorylation of CREB1 at Ser$^{133}$, STAT1 at Tyr$^{701}$, and STAT3 at Tyr$^{705}$. Relative phospho-protein levels under each condition were normalized to untransfected control cells (Ctrl) by setting this to 100 on the y-axis. Paired t tests were performed in Excel for each kinase knockdown condition compared to siCtrl or siCtrl with 24h DOX+NaB. P-values * ≤ 0.05, ** ≤ 0.01, and *** ≤ 0.001.

systems [68–74]. Thus, our results highlight the previously undescribed roles of ERBB2 and ERBB3 upstream of CREB1, STAT1 and STAT3 phosphorylation during KSHV reactivation.

## Discussion

Protein kinases are known to regulate numerous host and viral processes. Like other viruses, KSHV relies on virus-encoded [75] and host-encoded kinases for optimal lytic replication [19,21,22,76]. Some kinases regulate reactivation by altering LANA latent protein functions, lytic gene transcription and translation, and cell survival and proliferation [18–20,23,77–80]. Kinases with previously confirmed pro-lytic functions, including Pim-1/3 [19,23], MSK1/2 [18], and RSK/ERK [20,80,81], appear to act by regulation of latency-associated nuclear antigen (LANA) phosphorylation, activation of CREB1 lytic gene transcription factor, and cell proliferation and survival, respectively. Kinases with pro-latency roles include CDK6 which interacts with KSHV v-cyclin and nucleophosmin to restrict lytic replication [79], and AMPKα1 which restricts lytic replication following primary infection through an unknown mechanism [77]. While the role of kinases as regulators of KSHV reactivation has been established, the previously published screens have limitations. For example, kinase overexpression screens can result in artificial kinase catalytic activity, localization, or protein interactions; surveys of protein phosphorylation can implicate activated signaling pathways with substrates shared by many kinases; and proteome analyses can inform on kinase abundance but not on catalytic activity or function [18,23,24]. Additionally, screening of KIs can result in many off-target effects due to the inherent broad activity of these drugs [25]. The polypharmacology-based KiR kinome screening approach that we employed is not without limitations but takes a unique approach as compared to these previously published screens to build on our understanding of kinases and KSHV reactivation.

A key advantage of our method is the use of data characterizing the varying potencies of KIs for multiple targets to enable predictions of KIs and specific kinases that regulate reactivation. The KiR approach only requires experimental data from a subset of drugs (29 inhibitors), rather than testing all 427 KIs. As a result, the algorithms prioritize comprehensiveness over specificity, with the aim of extracting the maximum information on drug-target interactions from the original matrix of 427 compounds and their inhibitor effects on 371 kinases. The predictive capabilities of the set of 29 KIs have been assessed and validated in previous studies to (1) identify Fyn kinase as a mediator of cell migration in mesenchymal cancer cells [27], (2) identify PP121 and SC-1 as inhibitors of castration-resistance prostate cancer cell growth [82], and (3) identify a set of kinases and inhibitors that could suppress SARS-CoV-2 spike protein-mediated cytokine release in macrophages [30]. Here, our approach provides new insights into kinase regulation of KSHV latency and reactivation.

Using this KiR screening method, we were able to predict and validate both KIs and specific kinases that regulate KSHV latency and reactivation. Two of the top KIs included lestaurtinib, a broadly acting tyrosine KI and K252a, a staurosporine analog that also has broad activity. Neither of these drugs has been previously reported to affect the KSHV latent-to-lytic replication switch, although lestaurtinib restricts multiple stages of adenovirus replication in cell culture [83] and K252a impedes EBV lytic replication [84]. None of the six initially validated kinases, ERBB4, MKNK2, ITK, TEC, DSTYK, and FRK (Fig 3), had been specifically characterized previously as regulators of KSHV reactivation. The kinases that were predicted by the KiR screen but did not validate by siRNA knockdown included several MAPKs (MAP3K8, MAP2K2, MAP4K4), CAMK2G, and LRRK2, all of which have functional paralogs. It is possible that redundancies in the signaling pathways would require knocking down more than one of these kinases to reveal a reactivation phenotype. For example, CAMK2G forms complexes

with the other CAMK2 paralogs to generate a 12–14 subunit holoenzyme [85], which might retain function if only one member is depleted. Additionally, LRRK2 may be essential for cell survival as suggested by poor knockdown efficiencies and reduced cell viability in some experiments. Unlike the other kinases that did not validate from the screen, the PBK kinase does not have a paralog, but it has multifunctional roles in regulating cell cycle progression [86]. Therefore, knockdown of PBK may dysregulate opposing signaling pathways to disguise PBK functions during lytic replication. Lastly, knockdown of the CLK1 kinase was inefficient, so we could not evaluate its role in reactivation. While siRNA targeting CLK1 mRNA was published to knockdown both pre- and mature CLK1 mRNA relatively well, restoration of the depleted CLK1 RNA occurs rapidly under cell stress conditions [87] which may in part explain the poor knockdown in KSHV infected cells. Despite these limitations, siRNA-mediated knockdown of six kinases validated these as newly described regulators of KSHV reactivation.

The two validated screen hits with the strongest impact on reactivation, ERBB4 and MKNK2, are from kinase families with members that share high sequence homology and functional overlap. ERBB family members have well-characterized roles in cancer, cell proliferation, and cell survival, and they can form heterodimers to regulate the signaling pathways for these biological processes. MKNK1, like MKNK2, can phosphorylate the cap-binding protein eIF4E to regulate translation [88]. MKNK2 has also been shown to phosphorylate KSHV LANA latent protein in *in vitro* kinase assays [89]. To determine if these closely related kinases have pro-lytic functions like ERBB4 and MKNK2, we tested their effect on KSHV reactivation. Indeed, knocking down either ERBB2 or ERBB3 inhibited reactivation, even somewhat more strongly than ERBB4 (Fig 4D). In contrast, knocking down ERBB1 (Fig 5C and 5D) or MKNK1 (Fig 3B) promoted certain stages of lytic replication, suggesting that they positively contribute to latency maintenance. These studies illustrate that kinase paralogs with some shared signaling pathways can have parallel or counteracting roles in regulating the KSHV latent-to-lytic replication switch.

Published gene expression data for the ERBBs suggested that ERBB3 and ERBB4 expression is turned on following lytic induction (S6 Table, [36]). Consistent with these results, we observed switching on of ERBB3 and ERBB4 at the protein level and relatively constant ERBB1 and ERBB2 total protein (Fig 4A). ERBB2 is the preferred dimerization partner for the other ERBBs [39] and intriguingly remains constantly phosphorylated at Tyr$^{1248}$ and Tyr$^{1221/1222}$ under both latent and lytic induced conditions (Fig 4B and 4C), implying that ERBB2 may continuously reside in active dimers or larger oligomers [40,41]. On the other hand, ERBB1 Tyr$^{1173}$ phosphorylation, which is ERBB2-dependent during latency, decreased following lytic induction (Fig 6B). We speculate that increased ERBB3 and ERBB4 protein levels and decreased phospho-ERBB1 following lytic induction may indicate alterations in the composition and/or pairing of ERBB complexes. Curiously, catalytic inhibition of ERBB2 resulted in only a modest reduction in KSHV reactivation (Fig 4E) when compared to the more striking ERBB2 knockdown phenotype (Fig 4D). Likewise, the knockdown of the catalytically inert ERBB3 caused a pronounced reduction in KSHV reactivation (Fig 4D). These findings raise the possibility that in addition to catalytic roles of the ERBB family kinases, other biological features such as localization, dynamic transmembrane complexes, and cytoplasmic tail phosphorylation signatures that influence adaptor binding composition may all contribute to ERBB-dependent impacts on KSHV replicative switch.

We selected ERBB2 for further investigation because it plays a unique function in ERBB complexes as a dominant binding partner that drives prolonged signaling [39] and, in some cancers, behaves as an oncogene [90]. We reasoned that study of ERBB2 signaling may provide insight into how latency and/or lytic replication mediates KSHV-dependent oncogenesis. To determine which of the many signaling pathways regulated by the ERBB kinases are

ERBB2-dependent during latency and lytic replication, we assayed for the phosphorylation of key residues that are indicative of activation for a subset of signaling factors. The selected proteins included plasma membrane proteins involved in crosstalk with ERBB kinases, downstream signaling intermediates of ERBB kinases, and downstream substrates of the signaling intermediates (Fig 6A). Activation of plasma membrane receptors may contribute to the ERBB2 pro-lytic mechanism via receptor crosstalk pathways. For example, the MET oncogene can activate ERBB1 in some cancer cells and in others it is activated by ERBB1:ERBB2 heterodimers [44,46]. Also, signaling by PDGFRβ overlaps with ERBB signaling intermediates [47]. Our data show that MET signaling decreases with reactivation through an ERBB2-independent mechanism, while PDGFRβ phosphorylation was unchanged under all tested conditions (Fig 6C and S5A Fig). These findings do not support a role for MET or PDGFRβ as downstream factors of ERBB1:ERBB2 signaling.

From the investigation of signaling intermediates, we found evidence consistent with ERBB1:ERBB2 signaling during latency. In cells containing latent virus, AKT activation was reduced by ERBB2 depletion and other intracellular kinases, SRC and ERK1/2, showed similar although not statistically significant trends (Fig 7A). Contrary to the latent state, during lytic replication ERBB2 does not appear to activate these signaling intermediates. We also assayed for the role of JAK family kinases as these kinases are intermediates of ERBB signaling cascades and the JAK3 Inhibitor VI restricted reactivation (Fig 2D). The individual JAK family kinases did not have pro-latent or pro-lytic phenotypes under DOX-induced RTA plus NaB conditions (Fig 7C), but we did observe a moderate pro-latent phenotype for JAK1 under DOX-induced RTA alone conditions (S5C Fig). Therefore, JAK1 may also be activated by ERBB1:ERBB2 or another factor to promote latency. But none of the tested signaling intermediates likely contribute to the ERBB2-driven promotion of lytic replication.

The last category of factors that we tested are downstream substrates of the selected signaling intermediates, although we recognize the caveat that other factors may regulate these substrates. We found that ERBB2, and to some extent ERBB3, were required to fully activate CREB1 (Fig 8 and S7 Fig), a transcription factor that is activated by MSK1/2 and promotes KSHV lytic gene expression [18]. Our data do not clarify if CREB1 functions upstream of, in parallel with, or downstream of KSHV RTA. ERBB2 and ERBB3 also activated STAT1 and ERBB2 was necessary for STAT3 phosphorylation during lytic replication while ERBB4 knockdown had little effect on the phosphorylation of these transcription factors (Fig 8 and S7 Fig). Pro-latency roles for STAT1 and STAT3 have been described [91] as well as a pro-lytic function for STAT3 in efficient progeny production [63]. Additionally, STAT1 and STAT3 are phosphorylated by KSHV vIL-6 which may participate as a signaling intermediate between the ERBBs and the STATs to carryout vIL-6-dependent functions in promoting cell proliferation, angiogenesis and tumorigenesis [64–67]. ERBBs might also directly phosphorylate these STATs, as has been demonstrated *in vitro* [70,71]. Interestingly, in some cancer cells, STAT1 can negatively regulate ERBB2/Neu-dependent transformation [70]. If this is the case in KSHV infected cells, STAT1 may provide a negative feedback loop to dampen ERBB2 signaling while ERBB mediated activation of CREB1 initiates lytic gene expression and STAT3, coupled with vIL-6, may mediate angiogeneic and tumorigenic signaling [64,92]. Future investigation of these signaling axes may inform on whether or not the ERBBs regulate angiogenesis and cellular transformation in KSHV infected cells.

Our working model of ERBB signaling supported by these studies suggests that ERBB1 and ERBB2 coupled signaling activates AKT and likely other downstream intermediates to promote KSHV latency. The induction of lytic replication activates ERBB3 and ERBB4 expression, providing new binding partners, one of which (ERBB3) has high affinity for ERBB2 that likely competes with ERBB1 for ERBB2 binding. We propose in our model that turning on ERBB3

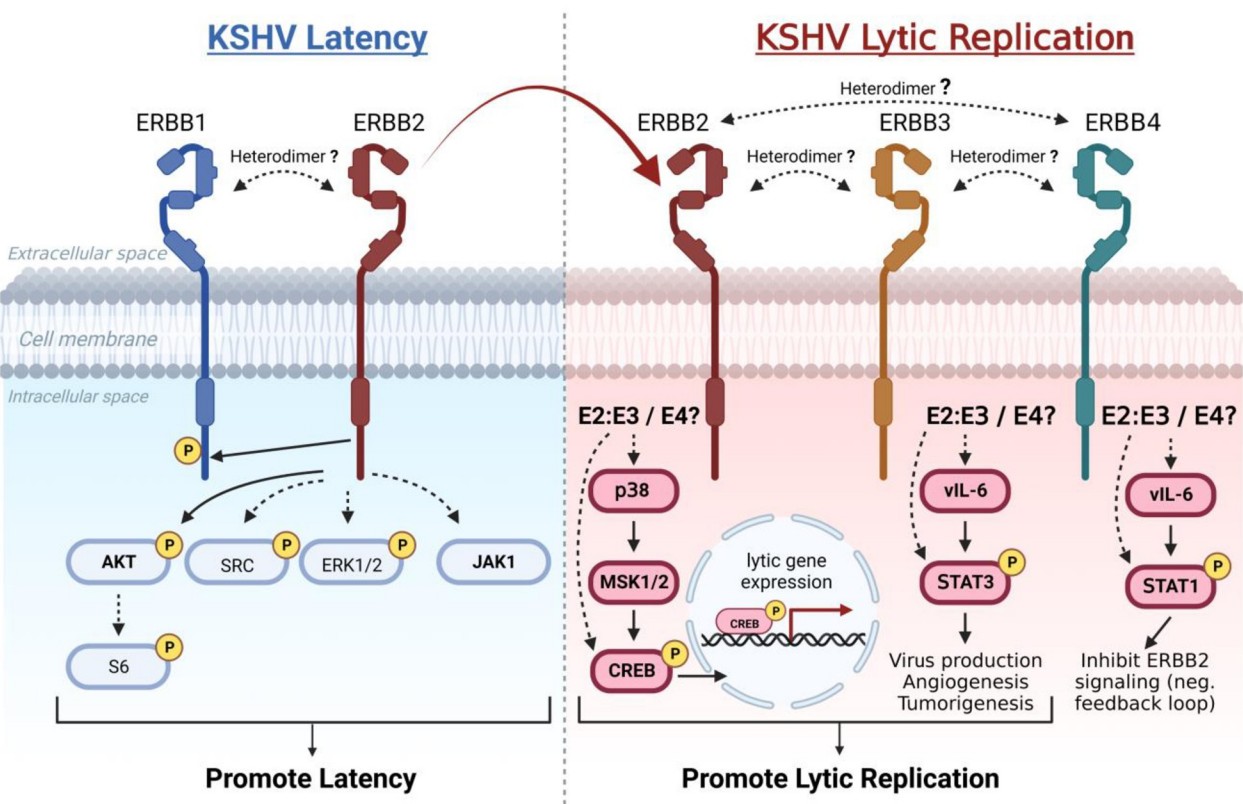

**Fig 9. Model of ERBB family kinase roles in regulating KSHV latent-to-lytic replication switch.** To promote the latent state (left), ERBB1:ERBB2 heterodimer signaling activates AKT and this trend applies to S6, SRC and ERK1/2 signaling intermediates. JAK1 in some contexts also promotes latency and may function downstream of the ERBB1:ERBB2 heterodimer. During lytic replication (right), this signaling is repressed as ERBB2 signals with newly expressed ERBB3, a switch which may be facilitated by high-affinity binding of ERBB2 to ERBB3, and ERBB2 and ERBB3-dependent signaling activates the enhancers of lytic replication. (ERBB2, E2; ERBB3, E3; ERBB4, E4; "Heterodimer" dotted lines represent proposed pairings; other dotted lines represent proposed signaling.) Diagrams in Fig 9 were generated using BioRender.

and ERBB4 expression during lytic replication contributes to KSHV replicative switch by causing changes to ERBB transmembrane complex composition and activation of CREB1, STAT1, and STAT3 lytic gene transcription factors, two of which are known to enhance KSHV lytic replicative cycle (Fig 9). ERBB mediated phosphorylation of these transcription factors may occur directly or indirectly through MSK1/2 in the case of CREB1, or vIL-6 for STAT1 and STAT3. These signaling pathways support the role of ERBBs in reactivation and imply a possible role in tumorigenesis and angiogenesis.

In addition to these pathways, other kinases identified in our screen may participate in downstream signaling from the ERBBs. For example, the ubiquitously expressed MKNKs are activated by p38, which is part of MAPK signaling pathways downstream of the ERBBs. The expression of another pro-lytic kinase validated from our screen, ITK, is induced following KSHV reactivation in iSLK cells (S6 Table, [36]). This and other TEC family kinases can interact with ERBB3 cytoplasmic tail following ERBB2-mediated phosphorylation [93,94] suggesting that TEC family kinases may function as signaling intermediates during the transition between latency and lytic replication. Continued dissection of the roles of kinases during KSHV lytic replication will provide insight into the overlapping or parallel pathways at work to coordinate this replicative switch.

Together, these experiments confirm the utility of a polypharmacology-based kinome screen to study KSHV reactivation regulators. The iSLK system provided a convenient cell line for these proof-of-principle studies with relatively high levels of induced-lytic replication, which is not achievable in most KSHV latent cell culture systems, though conducting this screening approach in other cell types and conditions is of interest. The translational potential of this research is most evident with the new connection identified between KSHV latent-to-lytic replication switch and ERBB signaling. Viruses encoding or overexpressing ERBB1 agonists demonstrate the role of ERBB1 signaling during viral life cycles including virus-mediated tumorigenesis [95–97]. Also, the ERBBs are well-studied regulators of tumorigenesis and have several targeted, FDA-approved therapies in use for some cancers [16,90,98,99]. Our investigation of kinase regulators revealed for the first-time counteracting roles of the ERBBs to coordinate the critical KSHV latent-to-lytic replication switch. Continued mechanistic probing of these factors including biochemical evidence of ERBB complex composition and ligand-mediated activation will enhance our understanding of the intricacies of this viral switch and application to other cell types and systems may inform on the therapeutic potential of targeting these kinases to affect KSHV-driven diseases.

## Materials and methods

### Cell culture

The renal carcinoma cell line (SLK) and doxycycline inducible RTA SLK (iSLK) cell line were a kind gift from Jae Jung (Cleveland Clinic) and Rolf Renne (University of Florida). The SLK, iSLK, 293T, and HeLa cell lines were maintained at 37˚C and 5% $CO_2$ atmosphere in Dulbecco's modified Eagle's medium (DMEM) supplemented with 10% NuSerum (Corning #355500) and 1% penicillin/streptomycin (Gibco #15140122). For iSLK cells, media was supplemented with 1 μg/ml puromycin (ThermoFisherScientific #BP2956100) and 250 μg/ml G418 (Sigma #A1720). After infection of iSLK cells, KSHV BAC16 or KSHV$^{LRI}$ episomes were maintained by addition of 1,000 μg/ml hygromycin B (Invitrogen #10687010) to the medium. All cell lines were confirmed to be mycoplasma negative using the MycoProbe kit (R&D Systems #CUL001B).

### Generation of KSHV$^{LRI}$ recombinant genome

The KSHV$^{LRI}$ recombinant genome was generated by recombining a dual lytic replication indicator cassette into the KSHV BAC16 genome. The KSHV BAC16 genome was generously provided by Jae Jung in the GS1783 *E. coli* strain [100]. The dual lytic replication indicator cassette was generated by PCR amplification of the KSHV *PAN* promoter (PrPAN) using BAC16 DNA as the template and primers 2580 and 2581 (S8 Table) and inserting the amplicon into the pcDNA3.1 V5-His-TOPO vector. The lytic replication indicators included a streptavidin-binding peptide fused to a truncated low nerve growth factor receptor (SBP-ΔLNGFR) and a nuclear-localized mCherry (mCherry-nls) separated by a P2A sequence. The SBP-ΔLNGFR was amplified from pJB-2045_CMV_SBPΔLNGFR (a gift from Jesse Bloom, Fred Hutchinson Cancer Center, [101]) with primers 2585 and 2586. The P2A-mCherry-nls was amplified from pEH_mCherry-NLS-TagRFP plasmid (a gift from by Emily Hatch, Fred Hutchinson Cancer Center, [102]) with primers 2600 and 2601. The PrPAN, SBP-ΔLNGFR, P2A-mCherry-nls DNA fragments were cloned into the pcDNA3.1 V5-His-TOPO vector at HindIII, KpnI/BamHI, and BamHI/NotI sites respectively.

To seamlessly introduce the lytic indicators into the KSHV BAC16, an additional I-SceI sequence [103] was cloned into the BamHI site between the SBP-ΔLNGFR and P2A sequences. Upstream of the I-SceI cleavage site, a 50bp DNA segment identical to the P2A sequence was

added to the forward primer. The I-SceI-KanR sequence was amplified from pEPKan-S2 (provided by Greg Smith (Northwestern University, [103]) with primers 2610 and 2611, where the forward primer contained the 50bp overlapping sequence and both primers contained BAMHI sites on the outer flanks. This DNA segment was cloned into the BamHI site of the pcDNA3.1 PrPAN-SBP-ΔLNGFR-P2A-mCherry-nls intermediate to make the pEQ1766 plasmid. Next, the PrPAN-SBP-ΔLNGFR-P2A-50bp-I-SceI-KanR-mCherry-nls gene cassette was amplified from pEQ1766 using primers 2617 and 2618. This cassette was inserted into the KSHV BAC16 genome containing GS1783 *E.coli* by seamless recombineering as described in [103,104].

The sequences of the KSHV[LRI] BAC and of BAC16 genomes were verified by Illumina deep sequencing. Briefly, 100 ng purified genomic DNA from a BAC16 or KSHV[LRI] clone were used to generate libraries using the KAPA HyperPlus kit and sequenced using an Illumina MiSeq. Reads were trimmed using Trimmomatic v0.39 and mapped to the human herpesvirus 8 strain JSC-1 clone BAC16 reference genome GQ994935.1 using Geneious read mapper [105,106]). Sequencing reads were deposited in NCBI BioProject PRJNA884721.

## Generation of stable KSHV[LRI] latently infected iSLK cells

KSHV latently infected iSLK cell lines were generated by first transfecting 293T cells with purified BAC16 or KSHV[LRI] DNA and subsequent co-culture with uninfected iSLK cells, as described by Jain et al. [107]. After selecting the transfected 293T cells with 100 μg/ml hygromycin, the virus was reactivated by adding 20 nM phorbol 12-myristate 13-acetate and 1 mM valproic acid to the medium. After 48-72h, the virus containing medium plus 8 μg/ml polybrene (Sigma #H9268) and the infected 293T cells were co-cultured with iSLK cells. Several days later, the co-culture media was changed to media supplemented with 500 μg/ml hygromycin, 1 μg/ml puromycin and 250 μg/ml G418 to select for KSHV latently infected iSLK cells. These cells were further selected with 1,000 μg/ml hygromycin and frozen in liquid nitrogen after two or three passages. Cells used for experiments were passaged less than 10 times.

## KSHV reactivation mCherry fluorescence assay

KSHV[LRI] latently infected iSLK cells with no prior treatment or those treated with siRNAs at 2-days post transfection were seeded into 96-well plates at $2.5 \times 10^4$ cells per well. The next day, cells were untreated, treated with 1 μg/ml DOX or treated with DOX and 1mM NaB for controls, or at the same time treated with kinase inhibitors for drug experiments. At 72 h post treatment, mCherry fluorescence object count per image was quantified using an Incucyte Imaging System (Sartorius) in two fields/well in at least triplicate wells. These technical replicates were averaged for each experiment and each biological replicate is represented as a dot for each bar graph condition. Cell viability was determined by percent cell confluence from phase images or GFP as measured by the Incucyte as an average per image. For the KiR screen kinase inhibitor sets and tofacitinib, data in which the cell viabilities were altered by > 30% as compared to the DMSO control were removed from analysis. For tucatinib or siRNA treated cells, data in which the cell viabilities were > 2.0 or 1.5 standard deviation, respectively (>31% reduction as compared to siCtrl cells under DOX plus NaB conditions) were removed from analysis. One replicate was removed for MKNK1, MKNK2, and LRRK2 siRNA mediated depletion experiments because of poor cell viability.

## Kinase inhibitor treatment

KSHV[LRI] latently infected iSLK cells were seeded at $2.5 \times 10^4$ cells per well into 96-well plates. The next day, the medium was replaced with medium containing the KI alone, KI plus 1μg/ml

DOX, or KI plus DOX and 1mM NaB. Control wells included medium with the vehicle control (DMSO at 0.2% or less) in place of the KI. Each treatment was conducted with 3 different wells as technical replicates. The initial training set of KIs described in Gujral et al [31] were tested at 2μM, 500nM, 125nM and 31nM concentrations. All 29 proprietary KIs for the initial training set ([31]; S2 Table) were tested once and 12 of these were tested twice and the two experiments were averaged. The validation set of KIs and tofacitinib (S2 Table) were tested at 2μM, 500nM, 125nM, and and additional 8μM for the KI validation set in two or three separate experiments. Tucatinib was tested at 7.8, 31, and 125nM in two or three separate experiments. All small molecule KIs for the initial training set were constituted in DMSO at 1mM stock solutions (0.2% DMSO for 2μM concentrations). The validation set KIs, tucatinib and tofacitinib were constituted in DMSO at 10mM or 4mM stock solutions (0.02% or 0.05% DMSO for 2μM concentrations).

## Kinase inhibitor Regularization (KiR) modeling

KiR models for KSHV reactivation were generated as previously described [31,35]. A training set of 29 KIs and validation set of 12 KIs were tested on KSHV$^{LRI}$ latently infected iSLK cells as described above, with the end result being a single response for 27 total KIs that represents the change in KSHV reactivation (as % DMSO control) at the profiled dose of the inhibitor. KI KSHV reactivation data was excluded for drugs in which the cell viability was altered > 30% or drugs with inconsistent dose responses. The kinase inhibition profiles of each inhibitor and the quantitative responses to those inhibitors were used as the explanatory and response variables, respectively, for elastic net regularized multiple linear regression models [108]. Custom R scripts (available at https://github.com/FredHutch/KiRNet-Public) employing the glmnet package were used to generate the final models [109]. Leave-one-out cross-validation (LOOCV) was used to select the optimal value for the penalty scaling factor λ. Models were computed for 11 evenly spaced values of α (the relative weighting between LASSO and Ridge regularization) ranging from 0 to 1.0 inclusive. These α values for the predicted KIs were converted into percentages with the control set to 100% in S3 Table. Kinases with positive coefficients in at least one of these models (with the exception of α = 0, which always has non-zero coefficients for every kinase) were considered hits (S4 Table).

## siRNA or plasmid transfections

KSHV$^{LRI}$ latently infected iSLK cells were seeded at 4x10$^5$ cells/well into a 6-well plate. The next day, cells were transfected with 100nM non-targeting control or individual kinase targeting On-TARGETplus SMARTPool siRNAs (Horizon Discovery Ltd; S9 Table) including four unique target sequences and 5μl RNAiMAX (ThermoFisher #13778150) in Opti-MEM (ThermoFisher #31985088) per well following manufacturer's instructions. For transfection with the JAK3 expressing plasmid, HeLa cells were seeded into a 6-well plate at 5x10$^5$ cells/well. The next day, cells were transfected with siRNAs as described above and 2 days following cell seeding, cells were transfected with 1.5μg pcDNA3.1-JAK3 and 3μl lipofectamine 2000 (ThermoFisher #11668027) per well in Opti-MEM following manufacturer's recommendations.

## Reverse transcription qPCR

Knockdown efficiencies of targeting siRNAs were evaluated by reverse transcription qPCR 3-days after siRNA transfection. For relative quantification of lytic gene expression, cells transfected with siRNAs were untreated or treated with 1 μg/ml DOX plus 1 mM NaB at 3-days post transfection and harvested at multiple time points following lytic induction; 8h for *RTA*, 24h for *mCherry* and *ORF10* and 48h for *K8.1*. The latent condition analysis for *RTA* (Fig 5A)

was harvested 3-days post transfection with siRNAs. Total RNA was extracted from cells following the RNAeasy Mini Kit (QIAGEN #74104) protocol and cDNA was synthesized from RNA samples according to the High-capacity RNA-to-cDNA (AppliedBiosystems #4387406) protocol. Quantitative PCR of cDNA samples were carried out using Power Sybr Green Master Mix (ThermoFisher #4367659) on a Bio-Rad CFX384 Real Time System C1000 Touch Thermal Cycler using primers listed in S8 Table. Relative mRNA levels were normalized to tubulin control and calculated using the ΔΔCt method for experimental conditions as compared to control conditions.

## Reverse-phase protein arrays (RPPA)

Control or siRNA treated KSHV$^{LRI}$ iSLK cells were harvested 4-days following siRNA transfection and 24h following KSHV lytic induction with 1 μg/ml DOX and 1 mM NaB for protein lysate microarray analysis. Sample preparations and protein array analyses were performed as detailed in Luckert et al. [110]. In brief, cells were rinsed then lysed in 50 mM Tris-HCl, 2% sodium dodecyl sulfate, 5% glycerol, 5 mM ethylenediaminetetraacetate, and 1 mM sodium fluoride, 1X Complete Protease Inhibitor Cocktail (1 tablet per 10 ml, Roche), 1X Pierce protease plus phosphatase inhibitor tablet (Thermo Scientific #A32959), 10 mM β-glycerol phosphate, 1 mM PMSF, 1 mM sodium orthovanadate, and 1 mM dithiothreitol. After filtering through a 0.2-μm filter plate, the lysates were printed onto a nitrocellulose-coated slide using Aushon 2470 arrayer. Primary antibodies listed in S10 Table were diluted 1:100 and incubated with slide for 24h on an orbital shaker at 4˚C. IRDye secondary antibodies listed in S10 Table were diluted to 1:1000 and incubated with slide for 1h, shaking at room temperature. The microarray slides were scanned in 680-nm and 800-nm channels with an Odyssey imager. Protein quantitation for siCtrl, uninduced KSHV$^{LRI}$ iSLK samples were set to 100 and samples were normalized to these controls. Six independent siRNA transfections were completed for the six RPPA experiments except for Fig 4C and S7 Fig in which only 3 independent experiments were performed.

## Statistics

Figures were generated using Graphpad Prism 8.0.1 software and statistical analyses were perfomed in Excel. Experimental conditions were compared to control conditions using either an unpaired or paired t test as indicated in the figure legend and significant p-values are noted in each figure (* ≤ 0.05; ** ≤ 0.01; *** ≤ 0.001). The numerical data used in all figures are included in S1 Data.

## JAK3 plasmid clone

The *JAK3* gene was amplified using primers 2636 and 2648 from the pDONR223-JAK3, a gift from William Hahn & David Root (Addgene plasmid # 23944; http://n2t.net/addgene:23944; RRID: Addgene_23944). The amplicon was cloned into the pcDNA3.1 V5-His-TOPO vector to generate the pEQ1797 plasmid. The sequence of the JAK3 insert in the resulting plasmid was verified by Sanger sequencing in the Fred Hutch Genomics Core.

## Immunoblot assays

Lysates of uninfected, KSHV$^{LRI}$ latently infected, or infected and 3-day lytically induced iSLK cells were separated on 8 or 10% polyacrylamide gels. Lysates containing 100,000–300,000 cells were loaded per lane for immunoblot assays except for Fig 4A and 4B, in which 45μg protein were loaded per lane. Total protein levels for lysates were quantified using Pierce BCA Protein

Assay Kit (ThermoFisherScientific #23225) following the manufactuer's specifications. For ERBB2 and paired actin blots, KSHV^LRI latently infected iSLK cells were harvested 2 days after siRNA transfection. For the JAK3 and paired actin control blots, HeLa cells were harvested 3 days after siRNA transfection and 2 days after pcDNA3.1-JAK3 transfection. For all gels, the protein was transferred onto a polyvinylidene difluoride (PVDF) membrane (Millipore), and proteins were detected by probing with specific antibodies (S10 Table) using the Western Star chemiluminescent detection system (Applied Biosystems) according to the manufacture's recommendations.

## Immunofluorescence assay

For LANA IFA, uninfected or KSHV^LRI latently infected iSLK cells were seeded into an 8-well plastic chamber slide (ThermoFisherScientific #177445) at $2x10^4$ cells/well. The next day, cells were fixed with 4% paraformaldehyde (Electron Microscopy Sciences #15710) in 1X PBS for 15m. Cells were permeabilized and blocked simultaneously for 20m with 1% Triton X-100 (Sigma #X100), 0.5% Tween20 (ThermoFisherScientific #J20605.AP), 3% BSA (Sigma #A7906) and 1X PBS (Gibco #14200075) solution. Nuclei were stained using DAPI containing mounting medium (VECTASHIELD #H-1200-10). For SBP-ΔLNGFR IFA, KSHV^LRI infected iSLK cells were seeded into a 24-well plate at $8.0x10^4$ cells/well. The next day, the cells were treated with DOX plus NaB. 5-days following treatment, cells were fixed as stated above, blocked with 3% BSA in 1X PBS, and incubated with streptavidin Fluor 680 conjugate (Invitrogen #S21378). Cells were imaged on a Leica Microsystems DM IL LED Fluorescent microscope with Leica Application Suite V4.12 software.

## Kinase activity profiles for JAK inhibitors

The kinase activity profiles for tofacitinib and JAK3 inhibitor VI (Fig 7B) were taken from the publicly available Kinhibition website (https://kinhibition.fredhutch.org/) [26]. Tofacitinib data indicates specific restriction of all JAK kinases and two other kinases, LRRK2 and PKN1. The JAK3 VI inhibitor restricts JAK3, TYK2 and 15 other kinases including Pim-1 and Pim-3 pro-lytic kinases and two kinases predicted from the kinome screen, CLK1 and MAP4K4. A heatmap of kinase activities during treatment with 500nM of either tofacitinib or JAK3 inhibitor VI was generated in GraphPad Prism 8.0.1.

## Supporting information

**S1 Fig. Expression of LANA and the lytic replication indicator SBP-ΔLNGFR from KSHV^LRI.** Uninfected iSLK cells or KSHV BAC16 or KSHV^LRI latently infected iSLK cells were (**A**) lysed and subjected to LANA immunoblotting or (**B**) analyzed by immunofluorescence for LANA puncta representing individual KSHV episomes. (**C**) KSHV BAC16 or KSHV^LRI latently infected iSLK cells were treated with 1 μg/ml DOX plus 1 mM NaB and incubated for 3-days before harvesting cells for immunoblot analysis of SBP-ΔLNGFR protein levels or (**D**) fixed and incubated with streptavidin-680 for imaging of SBP-ΔLNGFR on the plasma membrane of un-permeabilized cells.
(TIF)

**S2 Fig. Polypharmacology-based kinome screen in the absence of KSHV lytic inducing agents.** KSHV^LRI reactivation phenotypes were obtained from 20 of the 29 pre-selected KIs that had minimal changes (< 30%) to cell confluences as compared to the DMSO control as a measurement of cell viability and that demonstrated consistent dose responses curves. KSHV reactivation for control (black bar and dotted black line) and KI treatment (red bars) were

calculated as a percent of DOX plus NaB treated cells set to 100 from data in Fig 2C. In this graph, 1.0 represents ~0.2% of total cells and the dotted line represents spontaneous reactivation, ~0.02% of total cells or ~2 mCherry positive cells.
(TIF)

**S3 Fig. Kinase knockdown efficiencies for kinases validated from screen and MKNK1.** Knockdown efficiencies for siRNAs targeting specific cellular kinases were evaluated in KSHV$^{LRI}$ latently infected iSLK cells using RT-qPCR from total RNA harvested at 3-days post transfection with siRNAs. Relative mRNA levels were normalized to untransfected control cells (Ctrl) by setting this to 100 on the y-axis. The siCtrl transfected cells (siCtrl) were used to calculate the relative mRNA levels in the kinase-specific siRNA treated cells which are listed above each bar in the graphs. Data for ERBB1-4 are in S4 Fig.
(TIF)

**S4 Fig. Knockdown efficiencies and specificity for ERBB family members. (A)** Knockdown efficiency of ERBB2 targeting siRNA was evaluated by immunoblot for ERBB2 protein at 2 days following siRNA transfection of KSHV$^{LRI}$ latently infected iSLK cells. **(B)** Knockdown specificity for siRNAs targeting ERBB family kinases were evaluated in KSHV$^{LRI}$ latently infected iSLK cells using RT-qPCR from total RNA harvested at 3-days post transfection with siRNAs. Relative mRNA levels were normalized to untransfected control cells (Ctrl) by setting this to 100 on the y-axis. The siCtrl transfected cells (siCtrl) were used to calculate the relative mRNA levels in the kinase specific siRNA treated cells which are listed above each bar in the graphs.
(TIF)

**S5 Fig. Effects of ERBB2 and reactivation on phosphorylation of downstream signaling factors.** KSHV$^{LRI}$ latently infected iSLK cells were transfected with siRNA control or siRNAs targeting ERBB2 and then 3-days later untreated or treated with DOX plus NaB for 24h. Cells were harvested, and protein lysates were analyzed using a RPPA for phosphorylation of **(A)** plasma membrane receptor PDGFRβ at Tyr$^{1009}$ and **(B)** signaling intermediates (pan) PKC at Ser$^{660}$. Relative phospho-protein levels under each condition were normalized to untransfected control cells (Ctrl) by setting this to 100 on the y-axis. **(C)** Cell viability (grey bars) and KSHV reactivation (red bars) were measured for KSHV$^{LRI}$ latently infected iSLK cells transfected with siRNAs targeting individual JAK family kinases and 3-days later uninduced or treated with DOX alone for 72h. Control siRNA transfected cells treated with DOX (dotted black lines) were set to 100 and data for each condition was calculated as a percent of this control. Kinase knockdown efficiencies at 3-days following siRNA transfection were determined before addition of lytic inducing drugs and graphed in S6 Fig. For each knockdown, the efficiencies were averaged and listed below the corresponding kinase target as % KD. Identical to (A and B), quantification of phosphorylated **(D)** MARKS at Ser$^{152/156}$, **(E)** S6 at Ser$^{240/244}$, **(F)** NFκB P65 at Ser$^{536}$, and **(G)** total β-catenin protein were analyzed. Paired for (A,B,D-G) or unpaired (C) t tests were performed in Excel for each kinase knockdown condition compared to siCtrl or siCtrl with 24h DOX+NaB or for KI as compared to DMSO control. P-values $^*$ ≤ 0.05 and $^{**}$ ≤ 0.01.
(TIF)

**S6 Fig. Knockdown efficiencies and specificity for JAK family members. (A)** Knockdown specificity for siRNAs targeting JAK1, JAK2, JAK3 and TYK2 were evaluated using RT-qPCR from total RNA harvested at 3-days post transfection with siRNAs. Relative mRNA levels for JAK3 were below the level of detection for these samples. **(B)** HeLa cells were transfected with control or JAK3 targeting siRNA alone or in combination with a JAK3 expressing plasmid. Three days post transfection cells were harvested, and lysates were subjected to α-JAK3 and α-Actin

immunoblotting.
(TIF)

**S7 Fig. ERBB3-mediated signaling increases CREB1, STAT1 and STAT3 phosphorylation during lytic replication.** KSHV^LRI latently infected iSLK cells were transfected with siRNA control or siRNAs targeting ERBB2, ERBB3 or ERBB4 and then 3-days later untreated or treated with DOX plus NaB for 24h. Cells were harvested, and protein lysates were analyzed using a RPPA for phosphorylation of CREB1 at $Ser^{133}$, STAT1 at $Tyr^{701}$, and STAT3 at $Tyr^{705}$. Relative phospho-protein levels were normalized to untransfected control cells (Ctrl) by setting this to 100 on the y-axis. Paired t tests were performed in Excel for each kinase knockdown condition compared to siCtrl or siCtrl with 24h DOX+NaB. P-values * $\leq$ 0.05, ** $\leq$ 0.01, and *** $\leq$ 0.001.
(TIF)

**S1 Data. Numerical data for figures.** Excel file containing, in separate tabs, the numerical data for Figs 2B, 2D, 2E, S2, 3, S3, 4C, 4D, 4E, S4, 5, 6, 7A, 7C, S5, S6A, 8, S7.
(XLSX)

**S1 Table. Initial kinase inhibitor screen data for DOX plus NaB condition.**
(XLSX)

**S2 Table. List of kinase inhibitors and sources used for screening and validation.**
(XLSX)

**S3 Table. KiR-predicted kinase inhibitor responses for DOX plus NaB condition.**
(XLSX)

**S4 Table. KiR-predicted top 'informative' kinases for DOX plus NaB condition.**
(XLSX)

**S5 Table. Kinase inhibition profile of KI validation set used in DOX plus NaB screen.**
(XLSX)

**S6 Table. Kinase expression data from KSHV BAC16 infected iSLK cells.** The gene expression data normalized as fragments per kilobase of exon per million mapped fragments (FPKM) were taken from the published RNA-seq dataset GSE157275 [36] for the kinases predicted from the kinome screen and the additional ERBB and MKNK family members under latent and 48h lytic inducted with 50 μg/ml DOX plus 1.2 mM NaB conditions. No data was available for PKG2.
(XLSX)

**S7 Table. Kinase inhibiton profile for JAK3 Inhibitor VI.**
(XLSX)

**S8 Table. Primers.**
(XLSX)

**S9 Table. Dharmacon siRNA target and ID.**
(XLSX)

**S10 Table. Antibodies.**
(XLSX)

## Acknowledgments

The authors acknowledge the help of Drs. Alex Greninger and Nicole A. Lieberman from the University of Washington for help sequencing the KSHV^LRI recombinant.

## Author Contributions

**Conceptualization:** Annabel T. Olson, Behnam Nabet, Michael Lagunoff, Taranjit S. Gujral, Adam P. Geballe.

**Data curation:** Yuqi Kang, Songli Zhu, Taranjit S. Gujral.

**Formal analysis:** Annabel T. Olson, Yuqi Kang, Taranjit S. Gujral.

**Funding acquisition:** Behnam Nabet, Michael Lagunoff, Taranjit S. Gujral, Adam P. Geballe.

**Investigation:** Annabel T. Olson, Anushka M. Ladha, Chuan Bian Lim.

**Methodology:** Annabel T. Olson, Yuqi Kang, Taranjit S. Gujral, Adam P. Geballe.

**Project administration:** Annabel T. Olson, Taranjit S. Gujral, Adam P. Geballe.

**Resources:** Taranjit S. Gujral, Adam P. Geballe.

**Software:** Yuqi Kang, Taranjit S. Gujral.

**Supervision:** Annabel T. Olson, Michael Lagunoff, Taranjit S. Gujral, Adam P. Geballe.

**Validation:** Annabel T. Olson, Taranjit S. Gujral, Adam P. Geballe.

**Visualization:** Annabel T. Olson.

**Writing – original draft:** Annabel T. Olson, Adam P. Geballe.

**Writing – review & editing:** Annabel T. Olson, Yuqi Kang, Behnam Nabet, Michael Lagunoff, Taranjit S. Gujral, Adam P. Geballe.

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
