## [Decision Letter · Decision Letter 0]

7 Mar 2023

Dear Dr Geballe,

Thank you very much for submitting your manuscript "Polypharmacology-based kinome screen identifies new regulators of KSHV reactivation" for consideration at PLOS Pathogens. As with all papers reviewed by the journal, your manuscript was reviewed by members of the editorial board and by independent reviewers. The reviewers were in agreement that the work was novel and exciting and that the platform held promise for future viral latency/reactivation studies. Despite these positive responses, reviewers raised several concerns that will require additional experimentation, clarification and/or interpretation before the manuscript is suitable for publication in PLOS Pathogens. In light of the reviews (below this email), we would like to invite the resubmission of a significantly-revised version that takes into account the reviewers' comments.

Should you choose to revise and resubmit the manuscript, we ask that you respond to each of the major issues raised by the reviewers, paying particular attention to the following: the rationale behind the computer selection process and the selectivity of the inhibitors chosen; additional evidence for heterodimerization and switching; confirmation that inhibitors are not inadvertently influencing the magnitude of the Dox-induced RTA; pertaining to knockdown approaches, additional control/validation experiments; refinement of the mechanistic model.

Multiple minor issues specific to individual figures, associated text and discussion points were also raised. Please address each of these issues accordingly. Note that some of the concerns listed under major issues could be considered minor issues, but still require careful attention.

We cannot make any decision about publication until we have seen the revised manuscript and your response to the reviewers' comments. Your revised manuscript is also likely to be sent to reviewers for further evaluation.

Sincerely,

Ashlee V. Moses

Academic Editor

PLOS Pathogens

Blossom Damania

Section Editor

PLOS Pathogens

Kasturi Haldar

Editor-in-Chief

PLOS Pathogens

orcid.org/0000-0001-5065-158X

Michael Malim

Editor-in-Chief

PLOS Pathogens

orcid.org/0000-0002-7699-2064

Reviewer's Responses to Questions

**Part I - Summary**

Reviewer #1: Olson et al., employed a polypharmacology-based kinome screen to identify specific kinases that regulate KSHV reactivation. Results identified several kinases that enhance reactivation, whilst other promote the maintenance of latency. Further experiments focus on the potential mechanism of ERBB2 signalling.

This is a potentially interesting study examining the essential role of host cell kinases in the KSHV life cycle. It merits publication in the future, however, overall conclusions are unsubstantiated and require further validation and evidence, especially on mechanistic signalling claims.

Reviewer #2: (No Response)

Reviewer #3: In this manuscript, the authors use a polypharmacology-based kinome screen to search for kinases that support or suppress KSHV lytic reactivation, once triggered by inducible RTA expression and sodium butyrate (NaB) treatment. They engineer an elegant reporter system in iSLK cells, in which lytic gene expression is readout by a PrPAN driven mCherry reporter. This system is then used for a polypharmacology based kinome screen for enhancers or repressors of NaB/Dox induced reactivation. The idea is that kinase inhibitors have known sets of on/off target activity, and computational approaches can then be used to triangulate the relevant host cell kinase targets. The screen identified a number of host cell kinases, which were validated by chemical and siRNA approaches. Effects of TEK family kinase inhibition were noted, but the authors mostly focus on the interesting findings that ERBB2/3/4 inhibition impairs lytic replication, whereas ERBB1 was found to have a pro-latency role. Kinase inhibition alone was not sufficient to induce lytic replication, but functioned in the context of cells stimulated by Dox/NaB. Phosphoproteomic studies were used to characterize effects of iSLK lytic reactivation -/+ ERBB2 or downstream kinase blockade. Overall, this is an interesting study, which serves both as a proof-of-concept for the chemical screen approach in KSHV reactivation, which identified pro-latency/reactivation roles of ERBB kinases. The manuscript is well written and the figures are concisely and clearly presented. I would suggest addressing the following points prior to publication.

Reviewer #4: In the work described in this manuscript, Olson et al attempted to identify cellular kinase signaling pathways that regulate KSHV reactivation. They first engineered a reporter virus for easy cell-based assay; they next screened a small set of kinase inhibitors and then a larger set of inhibitors. Based on vaguely described Kinase Inhibitor Regularization (KiR) analyses, the authors selected 13 kinases for further analyses. SiRNA knock-down of the ERBB4(HER4) and MKNK2 (MNK2) noticeably reduced KSHV reactivation while knock-down of DSTYK (RIPK5), ITK, and TEC had moderate effects. The authors focused mostly on ERBB4 and MKNK2, and then examined the homologous kinases using similar approaches. Surprisingly, they found some of the homologues had the opposite effects. For example, they showed that ERBB2, ERBB3, ERBB4, and MKNK2 had pro-lytic/viral effects while ERBB1 and MKNK1 had anti-viral effects. They proposed that ERBB1 and ERBB2 form dimers in cells and the heterodimers between ERBB1 and other partners regulate KSHV reactivation. They also analyzed the downstream kinases and potent protein substrates. They provided evidence that CREB1 and JAKs might be involved. Overall, this chemical biology approach is novel and the results are interesting. However, the methodology was not described in enough detail. I had difficulty understanding how the top hit kinases were generated by the model, specifically how the experimental data were fitted with the known inhibitors profiles to produce the top hits. Furthermore, some of the experimental data were mostly correlative, inconclusive, and sometimes conflicted with each other.

**Part II – Major Issues: Key Experiments Required for Acceptance**

Reviewer #1: Throughout the paper is it pro-latent or anti-viral? More analysis is required to evidence this claim.

Line 137: More details would be helpful on how the computer selection occurred. Are the kinase selected upstream or downstream in a signalling cascade as any effects could be non-specific. How were these concentrations chosen? For example, all the concs used for lestaurtinib are above the IC50 for 3 major targeted kinases (JAK2, FLT3 and TrkA inhibitor (IC50 values are 0.9, 3 and < 25 nM, respectively). Considering this information, it is highly unusual no induction of apoptosis is seen. It needs to be shown that the inhibitors used are functioning, although I acknowledge this may not be possible for all the inhibitors used in the screen, confirming that the inhibitors are working at the concentrations used should be done for any inhibitors that are of downstream importance.

Line 213: The data is not strong enough to make this statement.

Line 276: Authors claim a downstream target of PKC is phosphorylated but earlier show PKC itself is not phosphorylated/activated (line 243).

Fig 2B: where are the dose response curves for the other kinases? Only 2 are shown - Some missing error bars.

FigS2: error bars. Lacking of detail in legend. How many cells were used to calculate this.

Fig3: The x axis for KD is unclear, are the authors using KDs that haven’t worked in this experiment? Large error bars. WB would be good to confirm KD.

FigS3: Confusing : what it is normalised to? as siCtrl are defined as 100% but often come up below 1.0 on the graph. There appears to be strong changes between ctrl and sictrl levels. Large variation in KDs, e.g. is RIPK7. ERBB4 KD data is missing compared to Fig3B. Additionally, functional KD of downstream signalling pathways should be given i.e. ERBB4 KD should lead to changes in PI3K, ERK and STAT signalling pathways. The KD of these genes alone gives no indication of the correct downstream effect. Additionally, FigS5: Why does ERBB2 KD not decrease p-MARCKS under normal conditions?

Fig4: The *** significance appears less significant than some of the * on the neighbouring graph with no explanation of the calculation. Also Fig 6: are these significant?

Line 240: It is unclear why there are so many repeats used for his experiment. Additionally, the decrease in p-AKT is not demonstrated to be relevant for the mechanism presented. To prove AKT signalling is needed, AKT should be directly targeted and the same effect should be seen (lytic reactivation, early and late gene and protein expression). Additionally, several AKT inhibitors can be found in table s2 but the predicted effect is very minor, contradicting the mechanism described here.

This comment can also be applied to ERK1/2 and SRC.

Line 268: This conclusion cannot be drawn from the data, it seems more likely that the JAK3 inhibitor VI is having an off-target effect. Additionally, no link between JAK1 and AKT/SRC/ERK1/2 is shown.

Line 288: Why was CREB chosen to be important, compared to the much larger change in STAT?

Additionally, there is not enough evidence linking ERBB2 and CREB. More validation is needed, such as use of inhibitors, D/N mutants etc.

Fig 9: Overall proposed schematic. This is rather over-interpreted from the data provided. The authors provide no evidence to show an ERBB1:ERBB2 interaction and that this interaction is disrupted in early lytic replication.

Reviewer #2: (No Response)

Reviewer #3: Initial screen could be confounded by effects on dox-induced RTA. Figure 3 should include a western blot to indicate effects on chemicals on RTA expression at an early timepoint of Dox induction (prior to downstream effects on RTA autoinduction via the lytic cycle). At a minimum, it would be important to show that the ERRB inhibitors do not simply function by perturbing the level of RTA driven by DOX the inducible system.

Would show levels of ERBB1-4 by qPCR or western blot for each knockdown, to also show the extent to which a given siRNA affects other closely related ERBB family members in Fig 4/S4.

Given the model, would present phosphokinase analysis for the other ERBB kinases as well, ERBB2-4. Data is presented only for ERBB1, but the data appears to be available for all ERBB kinases. Likewise, since a global phosphopeptide analysis was done for Fig 6-7, it would be nice if a volcano plot or some similar systematic analysis could be presented, to highlight the extent to which effects on ERBB phosphorylation and pathways are specific. For instance, based on the data we are shown, it is difficult to know whether many kinases are similarly affected by lytic reactivation, or whether the effects on ERBB are specific, which would nicely support the authors model.

Since this is important to the model, please provide more information about CREB1 Ser133 phosphorylation. Is this the key site that indicates CREB1 activation status? Were other CREB1 phosphorylation sites tested, or was this the only one on the array? Would likewise provide this information for STAT1 and STAT3.

To further support the model in Fig 9, would show that CREB1 knockdown reduces lytic gene expression in their model system. Why might CREB1 act apparently downstream of RTA? Would clarify in the model that CREB1 acts downstream of or at least in parallel with RTA (at present, the model doesn’t capture the fact that Dox/NaB were upstream activators).

Reviewer #4: 1. The Kinase Inhibitor Regularization (KiR) platform seems to make sense, but the method was not described in enough detail. Despite reading the manuscript a few times, I was still unable to figure out how exactly the top hits were generated, and therefore had a number of questions. For example, both top hits KI lestaurtinib and K252a are broad spectrum kinase inhibitors, what are the key experimental screening results that could single out ERBB4 and MKNK2 among their homologues? How many of the top hit kinase inhibitors could distinguish ERBB and MKNK homologues, or showed different inhibition profiles against ERBB homologues? Are there any more specific ERBB and MKNK inhibitors that can be used to lend more supportive evidence?

2. If indeed ERBB and MKNK families of kinases are important for KSHV reactivation, can the KiR model predict more specific inhibitors to these kinases? If yes, can those inhibitors be used as did the JAK inhibitors in this study? Because several downstream kinases are involved in the ERBBs signaling, can the KiR model predict more specific inhibitors for each downstream kinase? Use of those kinase inhibitors individually or in combinations to inhibit single or multiple pathways would provide a good validation and pinpoint which downstream kinase(s) is/are more critical.

3. For validation of the ERBB and MKNK kinases, SiRNA down experiments seem not enough. Constitutive active, kinase-dead, and dominant negative mutants should be used too.

4. The heterodimer model of ERBBs seems appealing but the model was not validated by any experiments.

5. Do ERBBs directly phosphorylate CREB1 at Ser133? If not, which downstream kinase would be responsible? Any data to support that kinase is involved?

6. Because ERBBs are EGF receptors, it would be interesting to examine whether EGF signaling can modulate KSHV reactivation.

**Part III – Minor Issues: Editorial and Data Presentation Modifications**

Reviewer #1: Statistical analysis is unclear with no method stated.

Reviewer #2: (No Response)

Reviewer #3: Typo in line 103: known to activat

Fig 2. Would enlarge * as they are quite small when viewed without enlarging the figure.

Line 145. Would better describe what is meant by 7 compounds being removed due to toxicity.

Line 277, would clarify “ERBB2 knock down lessened the induction but not to a statistically significant extent”. I think this is a typo, and rather than induction, phosphorylation is meant here.

Given concerns that there is very low expression of ITK, and therefore that 5-fold induction may still be very low levels of ITK, the question is raised of whether the single siRNAs used may have off-target effects. This could be addressed by demonstrating blotting ITK knockdown samples for both ITK and TEC.

Would clarify in the figure legends whether technical or biological replicates were used. Would add to the fig legend what was used as the load control for the qPCR analysis.

Would add to fig 6 legend how protein levels were normalized, given that this is the Y axis label.

Fig 7C. would add replicates to the tofacitinib experiments so that lack of significance can be properly assigned, given also the very large variance between the 2 replicates.

Would clarify an apparent paradox. It is stated in line 380 that during lytic replication ERBB2 does not appear to activate these signaling intermediates. And yet, ERBB2 was required to fully activate CREB1 (Fig 8).

Reviewer #4: 7. Although different concentrations of the inhibitors were used in the screening, it appeared that only the data from a single concentration (0.5uM) were used for the analysis (Table S2). Please explain.

8. More detailed legends or descriptions are needed for those tables, so readers can understand how the data were generated. The “Predicted KI responses at 0.5 uM” in the Table S2 and “Average Coefficient” in the Table S3 were not defined or explained. They are not self-explanatory to me.

PLOS authors have the option to publish the peer review history of their article (what does this mean?). If published, this will include your full peer review and any attached files.

Reviewer #1: No

Reviewer #2: No

Reviewer #3: No

Reviewer #4: No
---

## [Decision Letter · Decision Letter 1]

16 Aug 2023

Dear Dr. Geballe,

We are pleased to inform you that your manuscript 'Polypharmacology-based kinome screen identifies new regulators of KSHV reactivation' has been provisionally accepted for publication in PLOS Pathogens.

Best regards,

Ashlee V. Moses

Academic Editor

PLOS Pathogens

Blossom Damania

Section Editor

PLOS Pathogens

Kasturi Haldar

Editor-in-Chief

PLOS Pathogens

orcid.org/0000-0001-5065-158X

Michael Malim

Editor-in-Chief

PLOS Pathogens

orcid.org/0000-0002-7699-2064

All reviewers agree that this resubmission has addressed their concerns and satisfactorily responded to any additional comments. I have no additional concerns.

Reviewer Comments (if any, and for reference):

Reviewer's Responses to Questions

**Part I - Summary**

Reviewer #1: The authors have addressed the majority of issues raised by the Reviewers sufficiently. Additional experiments have supported the initial findings and strengthened the paper overall and text changes have further clarified issues. Whilst not all reviewers comments have been addressed fully the manuscript is now acceptable for publication and will be of general interest to the PLoS Pathogens readership.

A key issue raised by all reviewers was the over-interpretation of the ERBB2 switching model, with no direct evidence shown. Although not ideal the authors now provide additional data showing changes in ERBB-dependent signalling that have now been incorporated into a revised model.

Reviewer #2: (No Response)

Reviewer #3: The authors have addressed my concerns in their detailed and thoughtful rebuttal and revised manuscript. In my opinion, the revised manuscript is now suitable for publication in PLoS Pathogens.

Reviewer #4: The authors have made reasonable efforts to address my and other reviewers' comments. I would reserve my further comments for now.

**Part II – Major Issues: Key Experiments Required for Acceptance**

Reviewer #1: -

Reviewer #2: Major issues have been addressed

Reviewer #3: The major issues have been addressed in the rebuttal and revised manuscript.

Reviewer #4: (No Response)

**Part III – Minor Issues: Editorial and Data Presentation Modifications**

Reviewer #1: -

Reviewer #2: minor issues have been addressed

Reviewer #3: The minor issues have been addressed in the rebuttal and revised manuscript.

Reviewer #4: (No Response)

PLOS authors have the option to publish the peer review history of their article (what does this mean?). If published, this will include your full peer review and any attached files.

Reviewer #1: No

Reviewer #2: No

Reviewer #3: No

Reviewer #4: No

---

## [Editor Report · Acceptance letter]

31 Aug 2023

Dear Dr. Geballe,

We are delighted to inform you that your manuscript, "Polypharmacology-based kinome screen identifies new regulators of KSHV reactivation," has been formally accepted for publication in PLOS Pathogens.

Best regards,

Kasturi Haldar

Editor-in-Chief

PLOS Pathogens

orcid.org/0000-0001-5065-158X

Michael Malim

Editor-in-Chief

PLOS Pathogens

orcid.org/0000-0002-7699-2064